# The Pyrenean Platform for Observation of the Atmosphere: Site, long-term dataset and science

Marie Lothon[1], François Gheusi[1], Fabienne Lohou[1], Véronique Pont[1], Serge Soula[1], Corinne Jambert[1], Solène Derrien[1], Yannick Bezombes[1], Emmanuel Leclerc[1], Gilles Athier[1], Antoine Vial[1], Alban Philibert[4,1], Bernard Campistron[1], Frédérique Saïd[1], Jeroen Sonke[2], Julien Amestoy[5], Erwan Bargain[1], Pierre Bosser[6], Damien Boulanger[3], Guillaume Bret[1,†], Renaud Bodichon[9], Laurent Cabanas[1], Guylaine Canut[7], Jean-Bernard Estrampes[3], Eric Gardrat[1], Zaida Gomez Kuri[1], Jérémy Gueffier[1], Fabienne Guesdon[3], Morgan Lopez[10], Olivier Masson[11], Pierre-Yves Meslin[4], Yves Meyerfeld[1], Nicolas Pascal[8], Eric Pique[1], Michel Ramonet[10], Felix Starck[3], and Romain Vidal[11]

[1]LAERO, Université de Toulouse, CNRS, UT3, IRD, Toulouse, France
[2]Geosciences Environnement Toulouse, University of Toulouse, CNRS, UPS, Toulouse
[3]Observatoire Midi-Pyrénées, University of Toulouse, UPS, Toulouse
[4] Institut de Recherche en Astrophysique et Planétologie, Université de Toulouse, CNRS, UPS, France
[5]CEA, DAM, DIF, F-91297, Arpajon-Cédex, France
[6]Lab-STICC UMR 6285 CNRS/M3, ENSTA Bretagne/HOP, 29200 Brest, France
[7]CNRM-Université de Toulouse, Météo-France/CNRS, Toulouse, France
[8]AERIS/ICARE, CNRS/Université de Lille, Villeneuve d'Ascq, France
[9]ESPRI, Institut Pierre-Simon Laplace (IPSL), Paris, France
[10]Laboratoire des Sciences du Climat et de l'Environnement (LSCE), IPSL, CEA-CNRS-UVSQ, Université Paris-Saclay, Gif-sur-Yvette, France
[11]Institut de Radioprotection et de Sûreté Nucléaire (IRSN), PSE-ENV/SERPEN/LEREN, F-13115, Saint-Paul-lez-Durance, France
†deceased, 22 June 2016

**Correspondence:** Marie Lothon (marie.lothon@aero.obs-mip.fr)

**Abstract.**

The Pyrenean Platform for Observation of the Atmosphere (P2OA) is a coupled plain-mountain instrumented platform in southwest France. It is composed of two physical sites: The "Pic du Midi" mountain top observatory (2877 m a.s.l.) and the "Centre de Recherches Atmosphériques" (600 m a.s.l). Both sites are complementarily instrumented for the monitoring of climate-relevant variables and the study of meteorological processes in a mountainous region. The scientific topics covered by P2OA include surface-atmosphere interactions in heterogeneous landscape and complex terrain, physics and chemistry of atmospheric trace species at large scale, influence of local and regional-scale emissions and transport on the atmospheric composition, and transient luminous events above thunderstorms.

With a large number of instruments and a high hosting capacity, P2OA contributes to atmospheric sciences in (i) building long-term series of atmospheric observations, (ii) hosting experimental field campaigns and instrumental tests, (iii) educational training in atmospheric observation techniques.

In this context, P2OA is part of the French component of the Aerosol, Clouds and Trace gases Research InfraStructure (ACTRIS-Fr), contributes to the Integrated Carbon Observation System (ICOS) reserach infrastructure, and to several European or international networks.

Here we present the complete instrumentation of P2OA and the associated datasets, give a meteorological characterization of the platform, and illustrate the potential of P2OA and its dataset with past or ongoing studies and projects.

## 1 Introduction

The Pyrenean Platform for Observation of the Atmosphere (P2OA) is a ground-based facility devoted to research in atmospheric sciences. It is composed of two physical sites: The "Pic du Midi" mountain top observatory (2877 m a.s.l., hereafter PDM) and, 28 km apart in the plain, the "Centre de Recherches Atmosphériques" (600 m a.s.l, hereafter CRA).

It is one of the five national multi-instrumented sites of the National Institute of Universe Science (INSU) at the National Centre of Scientific Research (CNRS), devoted to the observation of the atmosphere, and one the foundation stones of the national research infrastructure ACTRIS-Fr, the French component of the Aerosol, Clouds and Trace gases Research InfraStructure (ACTRIS Pappalardo et al., 2018). INSU's other instrumented sites for observation of the atmosphere are:

– SIRTA, "Site Instrumental de Recherche par Télédétection Atmosphérique" (Haeffelin et al., 2005), close to Paris,

– CO-PDD, "Cézeaux-Aulnat-Opme-Puy De Dôme" (Baray et al., 2020), in the Auvergne massif, in Center of France,

– OPAR, "Observatoire de Physique de l'Atmosphère à La Réunion" (Baray et al., 2013), at La Réunion island,

– OHP, "Observatoire de Haute Provence", within a Mediterranean forest in south-Eastern France (see url Table C1).

P2OA thus belongs to a vast variety of ground-based observational platforms settled around the world, which gather on the same location a large number of complementary instruments for a comprehensive exploration of atmospheric processes. Ground-based observations are complementary to airborne or spaceborne remote sensing atmospheric measurements, since they offer the possibility to operate big, heavy, high-precision, or care-demanding instruments, more easily, or over a longer term, than on most mobile platforms. Sedentary ground-based atmospheric research stations can nowadays be found in almost all geographical environments on Earth:

– polar stations(e.g. Concordia in Antarctica, Argentini et al., 2005)

– maritime stations on isolated islands – often on top of high volcanoes, e.g. Mauna Loa in Hawaii (Keeling et al., 1976), Izaña in the Canary Islands (Gomez-Pelaez et al., 2019), Maïdo in the Reunion Island (Baray et al., 2013) – or on coastal places (Mace Head in Ireland (Milroy et al., 2012), Cape Grim in Tasmania (Chambers et al., 2016), Barbados Cloud Observatory (Stevens et al., 2016))

– continental stations, as developed below.

There are well known and widely used sites across the world like the Atmospheric Radiation Measurements (ARM) Southern Great Plains site in the USA (Mather and Voyles, 2013), or Cabauw (Bosveld et al., 2020) in the Netherlands.

ACTRIS (Pappalardo et al., 2018), a European research infrastructure focusing on climate-relevant short-lived atmospheric variables, aggregates no less than 79 ground-based observational platforms[1], among which a majority of continental stations. Part of these also belong to the dense network of atmospheric stations (around 50) of the Integrated Carbon Observation System (ICOS), the European research infrastructure devoted to greenhouse gases (GHG) monitoring[2]. Continental stations can be roughly categorized in flat terrain vs. mountain stations. Flat terrain stations are found in either rural (e.g. Cabauw Bosveld et al., 2020) or Hohenpeissenberg (Leuchner et al., 2015), peri-urban (SIRTA, Haeffelin et al., 2005) or urban environments (Qualair, in Paris, Ammoura et al., 2016), depending on scientific purposes or operational constraints. Only three sedentary mountain stations can be currently found in the Pyrenean area (Collaud Coen et al., 2018): P2OA is the only one located on the north side of the chain and under oceanic influence; Montsec is a mid-altitude station (1570 m a.s.l.) located about 100 km south-southeast of P2OA, on the Pyrenean southern flank (Pandolfi et al., 2014); the Montseny station (700 m a.s.l.) is settled on a Mediterranean coastal mountain in the aera of Barcelona (Pandolfi et al., 2011). The latter two stations are mainly devoted to aerosol and trace gases observations. P2OA is one of the few platforms which address a broader spectrum of atmospheric issues and observational techniques, with a large number of instruments and a significant hosting capacity of its infrastructure. For this, P2OA is involved in ACTRIS, ICOS and several other international networks devoted to atmospheric survey.

Despite operational difficulties, mountain sites have long been attractive for scientific experiments in atmospheric sciences (e.g. the proof of atmospheric pressure vertical gradient by Blaise Pascal at Puy de Dôme in 1648 – Baray et al., 2020) or climate monitoring (centennial meteorological data series at e.g. Puy de Dôme and Pic du Midi: Baray et al., 2020; Marenco et al., 1994, respectively). There are several reasons for such attractiveness:

– mountain meteorology as research topic in itself,

– need of cold or icing conditions,

– far horizontal visibility (e.g. for the optical observation of transient luminous events in the high atmosphere, see Section 3.5)

– thinner atmosphere above (e.g. for the study of cosmic rays interacting with the atmosphere, sun/moon photometer calibration , etc.),

– lesser influence of the continental atmospheric boundary layer, access to free tropospheric conditions,

– situation far from human activity.

A better knowledge in mountain meteorology, especially concerning the small-scale transport processes of atmospheric trace species, is an important challenge. Subgrid-scale vertical transport due to complex topography is not accounted for in global

---

[1]https://www.actris.eu/facilities/national-facilities
[2]https://www.icos-cp.eu/observations/atmosphere/stations

or regional-scale chemistry-transport models (Rotach et al., 2014; Bamberger et al., 2017). This is a major issue since complex topography cover more than 50% of Earth's continental surface, and small-scale vertical transport may affect, e.g., the global carbon balance at global scale (Rotach et al., 2014), but certainly also other species.

Composed of both a mountain-top and a lowland stations close to each other, with a rich instrumentation at CRA to profile the tropospheric dynamics above the site, and atmospheric composition measurement on both sites, the topographic and instrumental configuration at P2OA is especially suitable to address this question, as illustrated in Hulin et al. (2019).

The scientific topics covered by P2OA activity are based on the potential of the two sites in term of instrumentation, expertise and geographical embedding:

- atmospheric dynamics, surface-atmosphere interactions, planetary boundary layer in complex terrain and heterogeneous surface;

- physics and chemistry of atmospheric trace species at large scale, and their climate impact;

- influence of local and regional-scale emissions and transport processes on the atmospheric composition;

- atmospheric electricity, especially transient luminous events (TLEs);

- bio- and geochemical cycles in the environment.

P2OA contributes to atmospheric sciences in three major ways:

- building long-term observation series of climate-relevant variables from a large panel of complementary instruments;

- hosting experimental field campaigns dedicated to atmospheric process studies or tests of new observation techniques;

- educational training in the domain of atmospheric observation and instrumental techniques.

The goal of this article is to describe the platform, its instruments, and the associated long-term dataset. It also gives a meteorological characterization of P2OA, and reviews past or ongoing scientific studies based on P2OA infrastructure or data, in order to illustrate the platform potentials.

## 2  A plain/mountain double platform in the Pyrenees

P2OA is located on the north side of the Pyrenees, at similar distance as the crow flies from the Atlantic Ocean to the west (∼150 km) and the Mediterranean Sea to the east (∼200 km). Figure 1 shows the topography of the region and the location of the two sites. The Pyrenees main axis is mostly aligned along the west-east direction (more precisely, along the 300° - 120° axis). The highest peak is the Pico Aneto, at 3404 m a.s.l., and about 200 summits above 3000 m, most of them concentrated in the central part of the massif. On the French side, the main valleys are generally N-S orientated, transverse to the chain axis, while in Spain, the geometry is more complex with many E-W aligned sierras and valleys. The terrain lowers much more abruptly on the French side than on the Spanish side.

CRA site (43.128°N, 0.367°E) is located on the Plateau of Lannemezan, at about 600 m a. s. l., close to the exit of the Aure Valley. The first high ridge to the south, about 15 km away, reaches 1900 m a. s. l. (Bassia Mountain), that is 1300 m above the site. Several small hills and valleys start from the Plateau northward down to the Gers (district) cultivated plain. The Plateau of Lannemezan is covered by grasslands (at ∼30%, some of them wetlands or moors) and forests (at ∼30%, both deciduous trees and conifers), as well as crops to a lesser degree (mostly wheat and corn). CRA was built in the 1960's by Henri Dessens (1911-1971) for the study of convection, and has served atmospheric research since then.

PDM (42.936°N, 0.142°E – about 28 km southwest of CRA) at 2877 m a. s. l., is prominent for its astronomical observatory (Roudier et al., 2021), and its historical meteorological observations, which started in a heroic way in the early 1880's (Dessens and Bücher, 1995). PDM is nowadays easily accessible by cable car – a touristic resort being exploited at the summit. In the scope of P2OA, we consider only the instrumentation for atmospheric observation, installed on a dedicated platform in the summit buildings. PDM is the only high summit situated 15 km north of the chain of the highest peaks, the latter being concentrated along the French-Spanish border and water divide. This makes PDM a belvedere dominating the French plain, and probably justifies its equipment in the 1880's by Charles de Nansouty (1815-1895) and Célestin-Xavier Vaussenat (1831-1891). From a meteorological point of view, it is under the influence of air masses typically coming from the west, and mostly representative of the free troposphere composition (Section 6.1.2). The top of the peak is mostly made of rocks and pastures below.

The population in radii of 10 and 50 km around PDM amount to about 13,000 and 412,000 inhabitants, respectively, concentrated in two main cities: Pau (217,000 inhabitants) and Tarbes (110,000 inhabitants), situated 55 km and 30 km away to the northwest, respectively. A smaller town, Bagnères-de-Bigorre (7,000 inhabitants), is located 14 km north of PDM. Lannemezan (6,000 inhabitants) is located about 1.5 km east of CRA.

CRA and PDM are both equipped with complementary atmospheric instrumentation, described in the next section. Their infrastructures also allow the hosting of field experiments, education training and workshops, with lodging and meeting capacity. About 8 practical educational training sessions are hosted every year at P2OA, often organized as dedicated "micro-field campaign".

## 3   Instrumentation and data processing

Table 1 draws the list of the instruments in continuous operations at CRA and PDM. The main variables deduced from the measurements are indicated, as well as the algorithm used to obtain them from the raw data, and the network to which the instrument is connected. The list of instruments can also be found at https://p2oa.aeris-data.fr/instruments/, as well as a description of each instrument. In this section, we present more precisely those instruments and the associated algorithm. The corresponding observational networks are addressed in Section 4.

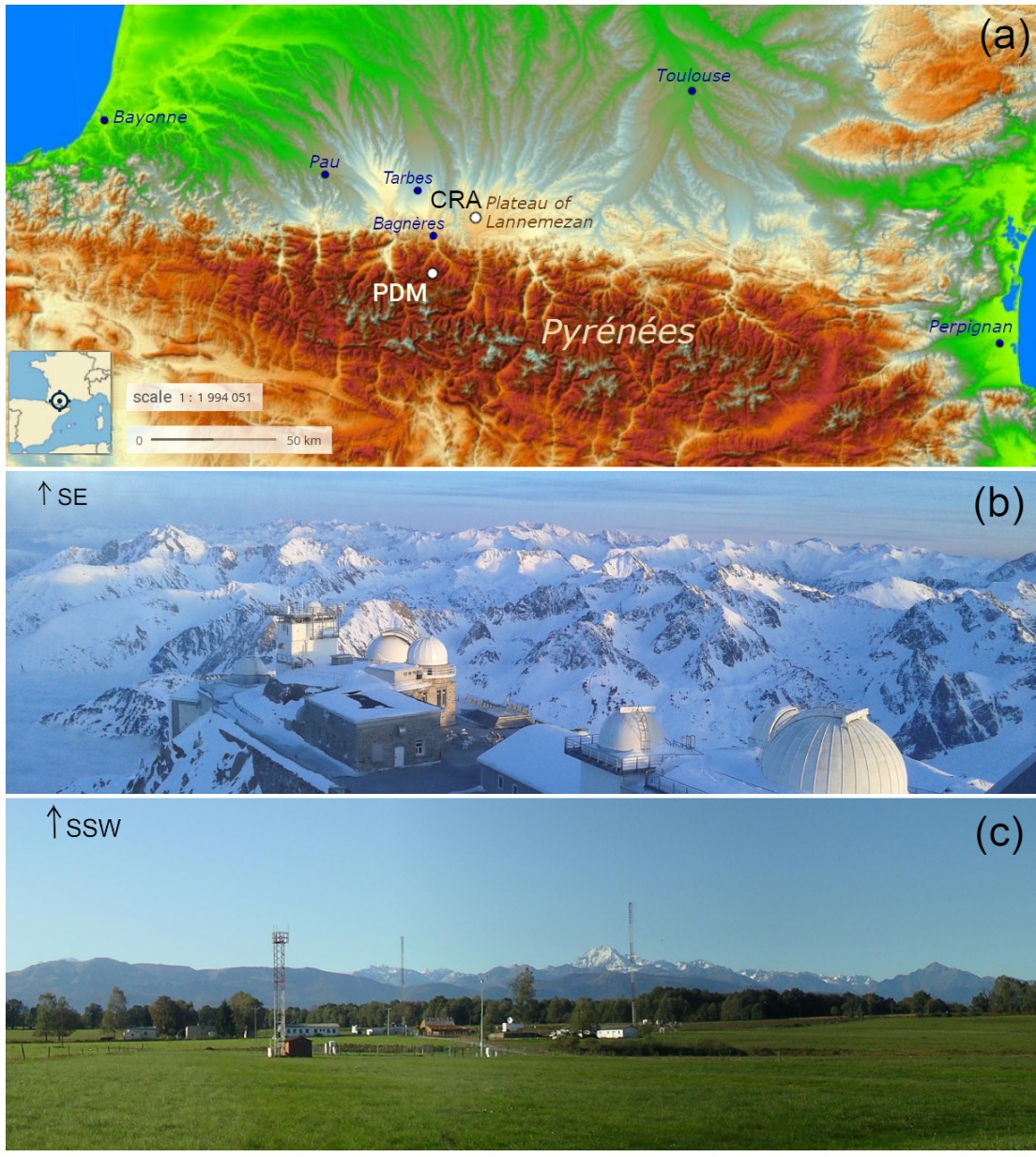

**Figure 1.** (a) Topography of Southwest France and location of the two P2OA sites, the "Pic du Midi" (PDM), and the "Centre de Recherches Atmospheriques" (CRA). ©IGN. Main towns are indicated in blue. Green colors from yellow-green to darker green represent altitudes from 0 to 200 m, yellow to red from 200 to 1600 m, red to brown from 1600 m to 2300 m, white and light grey for altitudes larger than 2300 m. (b) Picture of PDM observatory in winter. (c) Picture of CRA site. On this picture, PDM is visible with the snow on its top. In (a) and (b), the sight direction is indicated on the top left of the picture.

**Table 1.** List of permanent instruments at P2OA, with from left to right : the site where the instrument is located, type of instrument, main deduced variables, start time of the long term series, algorithm with which the raw data are processed (if appropriate), French network or French research infrastructure to which the instrument participates, international network or infrastructure. In this table, P, T and Hu design Pressure, Temperature and Humidity, respectively.

| Site | Instrument | Main variable(s) | Start time or period | Process algorithm | FR network or Res. Infrastr. | Int. network or Res. Infrastr. |
|---|---|---|---|---|---|---|
| CRA | 60-m instrumented tower | P, T, Hu, Wind, Radiation | 2010- | | ACTRIS-Fr | |
| | | Surface energy balance | 2010- | Eddy Pro | | |
| | 2-m flux station | Surface energy balance | 2014- | Eddy Pro | | |
| | Meteorological Station | P, T, Hu, Wind, Rain, Radiation | 1989- | | Météo-France | |
| | Meteorological Station | P, T, Hu, Wind, Rain, Radiation | 2016- | | StatIC | |
| | Instrumented pit | Soil temperature and moisture | 2019- | | ACTRIS-Fr | |
| | UHF radar wind profiler | Wind, TKE dissipation rate, BL height | 2010- | DESMAN | ACTRIS-Fr | E-PROFILE |
| | VHF radar wind profiler | Wind and tropopause height | 2001- | DESMAN | ACTRIS-Fr | E-PROFILE |
| | Full-Sky Camera - RAPACE | Cloud Cover | 2006- | ELIFAN | ACTRIS-Fr | |
| | Ceilometer CL61 | Cloud Base Height, $Z_i$, aerosol layers | 2022- | STRAT-Finder | ACTRIS-Fr | TOPROF |
| | GNSS Antenna | Integrated water vapor | 2011- | | Renag | |
| | DOBSON Spectrometer | $O_3$ total column | 2004- | O3-DOBSON | ACTRIS-Fr | NDACC |
| | Reactive Gaz Analyzers | $O_3$, CO, $NO_x$ | 2013- | | ACTRIS-Fr | |
| | Greenhouse-gas analyzer | $CH_4$, CO, $CO_2$ | 2019- | | ICOS-Fr | GAW, ICOS |
| | TLE Camera | TLE high sensitivity images | 2007- | UFOCapture | | |
| | Lightning Detector | Very Low Frequency radiation | 2008- | Time Of Arrival | | Linet |
| | High-volume sampler | Radionucleide activities | 2014- | | IRSN Opera | Ro5 |
| | Proportional counter | Gamma eq. dose. rate | 2020- | | IRSN Teleray | |
| PDM | Meteorological Station | P, T, Hu, Wind | 1882- | | Météo-France | |
| | Meteorological Station | P, T, Hu, Wind, Radiation | 2004- | | ACTRIS-Fr | |
| | Full-Sky Camera - EKO | Cloud Cover | 2017-2019 | ELIFAN | ACTRIS-Fr | |
| | GNSS Antenna | IWV | 2011- | | Renag | |
| | Aerosol filter sampling | EC, OC, inorganic ions | 2002- | | ACTRIS-Fr | |
| | Aethalometer | BC mass conc. | 2013- | | ACTRIS-Fr | ACTRIS-Eu |
| | Nephelometer | Part. scattering coeff. | 2013- | | ACTRIS-Fr | ACTRIS-Eu |
| | Optical Particle Sizer | Part. size distrib. (0.5-10 $\mu m$) | 2010-2021 | | ACTRIS-Fr | |
| | SMPS | Part. size distrib. (10 nm - 0.8 $\mu m$) | 2020- | | ACTRIS-Fr | ACTRIS-Eu |
| | Total suspended particles counter | Part. numb. conc. | 2013- | | ACTRIS-Fr | ACTRIS-Eu |
| | Reactive Gaz Analyzers | $O_3$, CO | 2001-, 2004- | | ACTRIS-Fr | GAW, ACTRIS-Eu |
| | Mercury Speciation | GEM, GOM, PBM | 2011-2014 | | | GMOS, iGOS4M |
| | Flask samples | $CO_2$, $CH_4$, CO, $N_2O$, $SF_6$ | 2001- | | ICOS-Fr | GAW, ICOS |
| | Greenhouse-gas analyzer | $CH_4$, CO, $CO_2$ | 2014- | | ICOS-Fr | GAW, ICOS |
| | Radon detector | $^{222}$Rn volumic activity | 2017- | | ICOS-Fr | |
| | High-volume sampler | Radionucleide activities | 2018- | | IRSN Opera | Ro5 |
| | Proportional counter | Gamma eq. dose. rate | 2009- | | IRSN Teleray | |
| | TLE Camera | TLE high sensitivity images | 2009- | UFoCapture | | |

## 3.1 Meteorological standard variables, surface energy balance, and atmospheric dynamics

Standard meteorological stations from Météo-France have been hosted at the two sites for many years. At PDM, it is actually an exceptionnal historical long time series which has been constituted, starting in 1882, maintained alternatively by Météo-France and Observatoire Midi-Pyrénées (the latter especially helping during World War II), and witness of the ongoing climate change (Dessens and Bücher, 1995). CRA and PDM stations measure the standard meteorological variables: temperature (2 m), humidity (2 m), pressure, wind (10 m), downward shortwave radiation (2 m) and precipitation at a time interval of 6 min. Another meteorological station is hosted from Infoclimat French participative science network(Garcelon et al., 2023), with all basic variables measured at 2 m[3].

The two sites are equipped with a GNSS antenna as part of the RENAG scientific GNSS network and are operational since 2011 (RESIF, 2017). The use of GNSS measurements for atmospheric water vapour measurements consists in estimating the propagation delay of GNSS signals in the atmosphere from the raw measurements. The integrated water vapour content can then be extracted from this delay (Bosser and Bock, 2021). The technique is well established and widely used in meteorology and climatology with an uncertainty of less than 1 kg m$^{-2}$ (Guerova et al., 2016). The raw GPS data acquired by the two antennas are routinely analysed with a latency of 14 days as part of the ACTRS-FR project. Propagation delays are estimated at a rate of 5 min and converted into integrated water vapour (IWV) using the methodology detailed in Hadad et al. (2018). The method uses hydrostatic delays and mean wet column temperature, which are calculated from 6-hourly ECMWF (European Centre for Medium Range Weather Forecasting) analyses and provided by Technische Universität Wien (Boehm et al., 2006).

One of the major instrumented structure of P2OA is the 60 m tower at CRA site, with 5 levels of slow meteorological measurements (temperature, humidity and wind, at 0.1 Hz at 2 and 15 m, and 1 Hz at 30, 45 and 60 m), and 3 levels of fast (10 Hz) measurements (temperature, humidity, wind at 30, 45 and 60 m). The four radiative components are also measured at 60 m with pyranometers (downward and upward shortwave flux) and pyrgeometers (downward and upward longwave flux). All the sensors installed on the 60 m tower are detailed in Table A1 in the Appendix. The 60 m tower is mainly surrounded by prairies, but with also small forests and crops in the vicinity, which more or less also contribute to the footprint, according to the wind and stability. Another flux station is installed at CRA site, at 2 m height within a prairie (fast measurements of temperature, humidity and wind at 2 m). Contrary and complementarily to the high tower which integrates a large heterogeneous landscape, this smaller tower measures flux at the scale of a land parcel. Fast measurements with sonic anemometers and hygrometers (Licor open path hygrometers) allow us to calculate the turbulent fluxes and moments, including sensible and latent heat, momentum flux, surface layer stability and other key turbulence indexes or scales, based on Monin-Obukhov theory. All terms of the surface energy balance between the earth surface and the atmosphere are thus measured with this instrumentation. The turbulent moments are calculated with the eddy-covariance method on 10-min and 30-min samples, with the EddyPro® Software (Version 6.2.0) from LI-COR Environmental [4]. The data process options have been discussed within AERIS and ACTRIS-FR, and are homogeneously applied to all ACTRIS-FR eddycovariance stations.

---

[3]https://www.infoclimat.fr/observations-meteo/temps-reel/campistrous-centre-de-recherches-atmospheriques/000CE.html#highlight=15

[4]www.licor.com/eddypro

Complementarily, soil temperature and moisture are measured at 6 levels into the ground (5, 10, 20, 30, 60 and 120 cm), as well as ground flux (5 cm).

Two radar wind profilers complementarily measure the wind vertical profiles from 150 m to 16 km above the ground: a Very High Frequency (VHF) radar and a Ultra High Frequency (UHF) radar. The UHF radar is a Degreane-Horizon PCL-1300, working at 1274 MHz, that is 23.5 cm wavelength, with 5 beams (one vertical beam and four oblique beams at 75° elevation). The sources of echoes are mainly the fluctuations of air refractive index, but also the hydrometeors when it rains, and the insects or birds in some conditions. The three components of the wind are deduced every 2 min with a 75 m vertical resolution in a low aquisition mode, or 150 m resolution in a high acquisition mode, based on the Doppler radial velocities of the 5 beams, with the velocity and turbulence volume processing technique (Campistron et al., 1991), based on the original Velocity Azimuth Display (VAD) technique (Browning and Wexler, 1968). This radar detects the top inversion of the convective boundary layer (Heo et al., 2003) and also allows us to estimate the turbulent kinetic energy dissipation rate (Jacoby-Koaly et al., 2002). An improved algorithm for the retrieval of the CBL structure has been developed by Philibert et al. (2024), which can detect thermal internal boundary layers and residual layers, in addition to the current CBL growth. Its maximum vertical coverage varies from 1000 m a. g. l. in dry winter days to 9 km in deep convection, but always includes the atmospheric boundary layer depth. Minimum height of measurements is close to 150 m a. g. l.

The VHF radar is a partly in-house-developed instrument, based on Degreane Horizon and TOMCO systems. It works at 45 MHz, that is 6.66 m wavelength, with 5 beams as well (one vertical beam and four oblique beams at 75° elevation). The sources of echoes come almost exclusively from the fluctuations of air refractive index, with parasite echoes from airplanes, which are filtered by the process algorithm. The three components of the wind are deduced with the same velocity volume processing technique as for the UHF, with a radial resolution of 375 m and a temporal resolution of 15 min. This radar can measure the wind profile from 1.5 km to 16 km a. g. l. It allows us to estimate the tropopause height, based on the local maximum of reflectivity generated by the specular echo that occurs at this strong inversion (Campistron et al., 1999; Kim et al., 2001)).

The algorithm which produces all the geophysical variables discussed before from the UHF and VHF radar is named DES-MAN (Jacoby-Koaly, 2000), and is related to a GIT-lab CeCIL license at AERIS data center. It is used as a homogeneous processing for several radar wind profilers settled in ACTRIS-FR sites.

All the instruments described previously are permanent instruments, operating continuously. In addition to this, P2OA is also equipped with mobile meteorological instrumentation, which can be used for educational training and for specific field experiments hosted at the site, or elsewhere: a MODEM radiosounding station, a tethered balloon with 5 meteorological probes, a flux station.

## 3.2 Clouds

Both sites are equipped with a total sky imager, which enables us to visualize the whole sky ($2\pi$ steradians) above each site. They routinely store the full sky images of the local cloud cover as seen from the ground.

At PDM, the camera is an EKO-SRF02 manufactured system operated between 2017 and 2019. At CRA, a home-made system (named RAPACE) is in operation (Lothon et al., 2019), which consists in the association of a digital camera and a large

angle lens (fish-eye). The latter is protected by a plexiglas dome and controlled by a thermostat which keeps the temperature nearby the lens around 10°C, in order to avoid condensation. RAPACE has been continuously operated at CRA since February 2006, with visible daytime and nightime images every 15 minutes (before February 2017) or 5 minutes (since then). During the night, a longer 15 s exposure time is used, for astronomy application. The acquisition frequency can be increased anytime

(up to 1 Hz) in the context of field experiments or specific demands, with the possibility of making movies.

An algorithm named ELIFAN has been developed (Lothon et al., 2019), in order to estimate the cloud cover from each image. It is based on red/blue ratio thresholding, both with an absolute or a differential (with a clear sky library) approach. All the pixels are thus attributed to either cloud, clear sky or 'uncertain'. Note that roughly 5% of the pixels are attributed as uncertain in partly cloudy images, which gives and estimation of ELIFAN uncertainty. ELIFAN has been generalized and

205 adapted to several other sky cameras like the EKO camera of the Pic du Midi, and other total-sky camera of ACTRIS-Fr infrastructure. It is associated with a Git-lab CECIL license.

Cloud base height is estimated from a ceilometer. From 2016 to 2019, a ceilometer Vaisala 25K was present at CRA site, giving estimates of cloud base height at three potential levels. Starting in April 2022, a new Vaisala CL61 ceilometer was installed, with improved capabilities: it can detect cloud base up to 16 km above ground, give information on the vertical

distribution of aerosols derived from the backscatter profiles, and also measure the linear depolarization ratio. The latter enables us to distinguish spherical particles from dissymetrical particles, like liquid water from ice within clouds or precipitation, and types of aerosols. This ceilometer also enables to estimate the boundary layer top. Through the participation to E-PROFILE, the StratFinder (Kotthaus et al., 2020) algorithm will be applied in the future, which supplies different interfaces from the vertical structure, including the convective boundary layer top.

The knowledge of cloud occurrence at PDM summit, that is whether it is immersed within cloud or not, is relevant for in situ measurements of soluble gas concentrations or aerosol properties. Since April 2022, a binary index indicating cloudy conditions at PDM (called the "in-cloud index") has been derived from images taken every 5 minutes with a webcam showing details of the summit platform (from a few meters to 150 m away from the camera), as well as the background mountain landscape. (A similar approach had been developed by Baray et al., 2019, for the Puy de Dôme observation platform.) The

algorithm is based on edge detection using a Canny filter on several areas in the image, selected at various distances. Areas with sharp contours are considered free of cloud, while blurred contours are considered as the signature of fog between the scene and the camera. These various pieces of information are then merged using a fuzzy logic algorithm finally returning 0 (false) as value for clear air, or 1 (true) for the presence of cloud.

## 3.3 Atmospheric trace gases

### 3.3.1 Reactive gases

*$O_3$ and CO*

In the decades following Schönbein's discovery of ozone ($O_3$) in 1839 and the characterization of the molecule (trioxygen) by Soret in 1865, the question arose of the presence of this gas in the atmosphere. Among the very first historical tropospheric

ozone measurements, some were conducted at the Sencours station – settled at a saddle-point 500 m below PDM summit – as early as 1874, then at the new summit station from 1881 to 1909, using the Schönbein's paper method (Marenco et al., 1994). Modern measurement series by UV-absorption analyzer were conducted at PDM in 1982 (Marenco, 1986), in 1990-1993 (Marenco et al., 1994), and then continuously since 2001 to present. Compilation of all those data series revealed an increase of tropospheric ozone by a factor 5 from the 1900's to the early 1990's (Marenco et al., 1994) but a stabilization around 45 nmol mol$^{-1}$ since then (Chevalier et al., 2007). As a chemical precursor of tropospheric ozone, carbon monoxide (CO) was also measured during the 1982 campaign (Marenco, 1986), and continuously since 2004 by trace-level IR-absorption analyzer (Gheusi et al., 2011). Technical details on the $O_3$ and CO instruments, as well as uncertainty calculations are given in Gheusi et al. (2011). The World Meteorological Organization (WMO) Global Atmospheric Watch (GAW) standard operation procedures (GAW, 2013, 2010, for $O_3$ and CO, respectively) are followed at PDM.

The stratospheric ozone is also monitored at P2OA, with a Dobson UV spectrometer which measures the total ozone column, by use of the O3EDOBSON[5] algorithm.

*Mercury*

In May 2011, automated atmospheric mercury speciation sensors were installed at PDM for gaseous elemental mercury, gaseous oxidized mercury and particulate-bound mercury. The instrumentation has been operated until the end of 2014 by the Geosciences Environment Toulouse research institute, and was composed of a Tekran ambient mercury vapor analyzer (model 2537A/B), a mercury speciation unit (model 1130) and a particulate mercury unit (model 1135). PDM joined the Global Mercury Observation System (GMOS, see Table C1) project as an external site in 2012.

### 3.3.2 Greenhouse gases

Continuous measurements of the two main Greenhouse gases, carbon dioxide ($CO_2$) and methane ($CH_4$), as well as carbon monoxide (ancillary of CH4 measurements) are conducted in the framework of the national RAMCES/ICOS network since May 2014 at PDM[6], and April 2019 at CRA[7]. These measurements are made on both sites by means of Picarro G24 analyzers based on cavity ring-down spectroscopy (Yver-Kwok et al., 2015). The applied QA/QC protocol (tracability of data and calibration chain; regular human-eye data check; etc.) is defined by the European programme ICOS.

Prior to these continuous Greenhouse gases measurements, air flask sampling has been made weekly since 2002 at PDM. Flasks were then sent to the RAMCES service at LSCE for laboratory analyses of $CO_2$ (with speciation of isotopes $^{13}$C et $^{18}$O), $CH_4$, CO, $N_2O$ and $SF_6$.

### 3.3.3 Radon and radioactivity

Radon is an inert radioactive gas emitted from ice-free soils with a half-life of 3.8 days, making it a useful tracer of the atmospheric boundary layer dynamics, and thus a reliable tracer to discriminate between air masses recently influenced by the

---

[5]http://www.o3soft.eu/

[6]https://icos-atc.lsce.ipsl.fr/panelboard/PDM

[7]https://icos-atc.lsce.ipsl.fr/panelboard/CRA

continental boundary layer, and the free troposphere (Chambers et al., 2013). A highly-sensitive radon monitor manufactured by the Australian Nuclear Science and Technology Organisation (model 1500L; Whittlestone and Zahorowski, 1998) is in operation at PDM since October 2017.

A continuous monitoring of radon concentrations has also been performed at three different heights of the CRA 60 m tower (at 1 m, 30 m, 60 m), with AlphaGuards ionization chambers, since 2018 (Amestoy, 2021). Atmospheric radon is also monitored by gamma-ray spectroscopy, using a RSX-5 NaI spectrometer mounted on the same tower, and soil concentrations of radon are measured by means of three Barasol probes (Amestoy et al., 2021).

P2OA hosts 2 of the 50 national aerosol samplers for the monitoring of atmospheric radioactivity, as part of the Permanent Observatory of Atmospheric Radioactivity (IRSN OPERA network). A very high flow rate sampler ($900 \text{ m}^3 \text{ h}^{-1}$, before temperature and pressure corrections) equips PDM, making it the most sensitive altitude station in terms of trace-level radionuclide measurements. At both P2OA sites, samples are taken on a weekly basis. Both samplers make it possible to detect naturally occurring radionuclides such as cosmogenic ones (e.g., $^7$Be, $^{22}$Na) as well as any long-lived artificial radionuclides (e.g., $^{137}$Cs) resulting from past nuclear tests or shorter-lived ones which could indicate a nuclear accident release. Results can be downloaded on the https://www.mesure-radioactivite.fr/#/ website. These devices closely complement the high-frequency measurements of automatic alert probes (TELERAY network) which would react instantly in the event of high contamination.

## 3.4   Physical, chemical and optical properties of aerosols

At PDM, the chemical composition of aerosols has been measured since 2002. Weekly filter samples are taken from a pumped volume of air of approximately 400 cubic meters. They are then analyzed to provide the concentrations of elemental carbon (EC) and organic carbon (OC) through a thermo-optical analytical technique (EUSAAR protocol; Cavalli et al., 2010) on the one hand, and the concentrations of major inorganic ions (calcium, magnesium, sulfate, nitrate, potassium, chlorine, ammonium) through an analysis by ion chromatography on the other hand.

Aerosol optical properties have been measured since 2013. Scattering measurements were implemented for one wavelength from 2013 to 2018 (Nephelometer ECOTECH M9003) and then for three wavelengths from 2018 to present (Nephelometer ECOTECH AURORA 3000). Aerosol absorption properties were measured first (2013 -2017) for one wavelength and then for seven wavelengths (2017 - present) with a Magee AE16 and then an AE 33 Aethalometer, respectively.

The total aerosol number concentration has been measured since 2008 by a condensation particle counter (TSI CPC 3010) until 2020 and then by a CPC 3750 thereafter.

Finally, the aerosol size distribution measurement was initially performed only in the coarse mode (0.5 -10 $\mu$m) from 2010 to 2021 thanks to the implementation of an Optical Particle Sizer (TSI OPS 3330), then from 2020 the aerosol size distribution from 10nm to 0.8 $\mu$m has been performed by the implementation of a Size Mobillity Particle Spectrometer (model 4S from Paolo Vilani société 4S).

### 3.5 Transient Luminous Events optical observations

Two low-light Watec 902H cameras (minimum illumination of $10^{-4}$ lux) are installed at the two P2OA sites. They have a field of view of 31° and are oriented to the storm with a pan-tilt unit that can be remotely controlled via the Internet. Thus, the observations of Transient Luminous Events (TLEs) can be made in night conditions, above thunderstorms in a range of about 800 km around each site, at the altitudes of these phenomena (between 30 km and 90 km), and on alert according to the meteorological conditions. The cameras operate in a triggered mode provided by the UFOCaptureV2 software[8] to capture luminous events with brightness above a given threshold. They provide videos with a time resolution of 25 frames (or 50 interlaced fields) per second, which corresponds to a time resolution of 20 milliseconds. The azimuth and elevation of the sprite events are determined with the software "Cartes du Ciel" (SkyCharts). The methodology for the analysis of the video imagery, the time and space synchronization with other data about the storm activity, especially the lightning flashes associated with the TLEs, the event terminology, are explained in Soula et al. (2017).

## 4 Data dissemination

### 4.1 Infrastructures and Networks

Most of the permanent instruments at P2OA participate to national or international atmospheric monitoring networks. Those are indicated in Table 1 for each instrument. All networks or infrastructures pursue common objectives: sharing expertise on improving data quality and standardizing the observing systems, procedures, data bases, data processing, and data dissemination. They also favour the research and development, educational training, technical assistance and production of essential output information to end-users.

#### 4.1.1 ACTRIS

P2OA is intrinsically linked with the national ACTRIS-France infrastructure – (hereafter ACTRIS-Fr), the French component of the European ACTRIS infrastructure (Pappalardo et al., 2018). ACTRIS is a distributed infrastructure in support of research on climate and air quality, for a better understanding of the evolution of processes and atmospheric composition. It supplies information on variability of climate-relevant reactive species from multiple observational and exploration platforms. ACTRIS-Fr has a larger scope than ACTRIS-ERIC (European Research Infrastructure Consortium, also ACTRIS-Eu in Table 1), by also including fundamental variables of climate and meteorology: atmospheric dynamics and surface/atmosphere heat fluxes. The plain involvment of P2OA within ACTRIS-Fr explains why so many instruments participate to this infrastructure (see Table 1). Only aerosols and trace gas continuous monitoring made at PDM participate so far to the European scale of the infrastructure, with involvment in two of the six ACTRIS topical centers: the European Center for Aerosol Calibration and Characterization, and the Center for Reactive Trace Gases In Situ Measurements.

---

[8]https://sonotaco.com/soft/e_index.html

By construction, ACTRIS-Fr is thus a convergence point of many networks, which is beneficial in both ways: the infrastructure helps the involved sites in maintaining there instrumentation and monitoring, in the data dissemination, brings a national scientific research dynamics, etc... and the network brings specific scientific questions, dynamics and tools from a european or international community, etc..

### 4.1.2 ICOS-ATC

Greenhouse gases measured at both PDM and CRA participate to the Integrated Carbon Observation System - Atmospheric Thematic Center (ICOS-ATC) at the European scale (Heiskanen et al., 2022). ICOS produces standardised long-term observations to understand the carbon cycle and to monitor the greenhouse gases, for a better understanding of climate change and its impacts. Atmospheric measurements of $CO_2$ and $CH_4$ at both PDM and CRA contribute to the European Obspack compilation updated once a year. The data are available on the ICOS Carbon Portal (https://doi.org/10.18160/PEKQ-M4T1 and https://doi.org/10.18160/9CQ4-W69K respectively for $CO_2$ and $CH_4$).

### 4.1.3 Météo-France

The two meteorological stations of P2OA managed by Météo-France – the French national meteorological service – are part of its synoptic network, providing operational observation data for assimilation by the Numerical Weather Prediction (NWP) models ARPEGE and AROME, and for climate monitoring.

### 4.1.4 E-PROFILE

The two wind profilers and the ceilometer participate to E-PROFILE, a program of the European network of National Meteorological Services, EUMETNET (see Table C1). E-PROFILE is part of the EUMETNET Composite Observing System, managing the European networks of radar wind profilers, lidars and ceilometers for the monitoring of vertical profiles of wind and aerosols. Near real time data of the two P2OA wind profilers are sent every 30 min to the network, with constraints of timeliness for the Global Forecast System to assimilate the data on time. Monthly statistics monitoring of the model-observation departure is supplied by the network and by Météo-France.

### 4.1.5 GAW

The Global Atmosphere Watch (GAW) programme (see Table C1) of the World Meteorological Organization (WMO) aims at improving the global understanding of atmospheric composition, and coordinates the collection of high-quality atmospheric composition observations from stations all over the world. Owing to the remote character of the site and the panel of long-term observations conducted there, PDM has been accepted as a GAW Regional Station since 2018.

Beforehand, PDM had already contributed to GAW databases for many years:

– to the World Data Center for Greehouse Gases (see Table C1), hosted by the Japan Meteorological Agency) since 2001,

- to the World Ozone and Ultraviolet radiation Data Center (see Table C1), hosted by the Canadian government since 2004 (see the NDACC section below),

- to the World Data Center for Reactive Gases, hosted at the Norwegian Institute for Air Research on the so-called 'EBAS' system (see Table C1) since 2007 (ozone),

- to the World Data Center for Aerosols since 2013.

### 4.1.6 NDACC

The Network for the Detection of Atmospheric Composition Change (NDACC, see Table C1) is an international collabora-
355 tion and a worldwide network of ground-based stations for remote-sensing observation of (mainly) water-vapor and ozone in the whole atmosphere (from the troposphere to the mesosphere). At the national scale, observations are coordinated by the NDACC-France service. P2OA is contributing to the NDACC database (and in the same time to the GAW World Ozone and Ultraviolet Radiation database) with data from a Dobson UV sprectrometer (historical instrument #49) operated at CRA since 2004.

## 4.2 Data and physical access policies

### 4.2.1 Data policy

The data are available from P2OA web portal (https://p2oa.aeris-data.fr/data/), but also accessible from the ACTRIS-Fr web portal (https://www.actris.fr/actris-fr-data-centre/) and AERIS data catalogue (https://www.aeris-data.fr/en/catalogue-en/) for the concerned variables, with no difference on the pointed dataset according to the portal used. Long-term observation data
collected on any P2OA site, are ruled by P2OA data policy which is available here: https://p2oa.aeris-data.fr/p2oa-data-policy/. The data policy follows ACTRIS-Fr data policy for any variable laying in the contour or ACTRIS-Fr.

The main spirit of P2OA and ACTRIS-Fr data policies is to offer free and unlimited access to P2OA data.

The users agree to contact the concerned local principal investigators to offer an appropriate level of acknowledgement or collaboration. In all cases, publications using P2OA long-term observation data should include the acknowledgement formula
proposed in the data policy (or an appropriate adaptation of it).

### 4.2.2 Site physical access – Hosting policy

Observation data collected on P2OA sites in the context of temporary campaigns are not concerned by the data policy, but by the P2OA hosting policy, available at this address: https://p2oa.aeris-data.fr/physical-access-form/.

Beyond the respect of this policy, the temporary users of P2OA experimental sites are beforehand invited to fill a physical
access form where the technical needs of their experiment should be described in sufficient detail. Required at least one month in advance, this form allows P2OA staff to assess the feasibility of the project and prepare best technical conditions for the experiment, in close interaction with the hosted research team.

Again, the users are requested to agree with the local principal investigators on the appropriate level of collaboration. As minimum requirement, the acknowledgement formula mentionning P2OA should appear in any publication.

### 4.3 The ReOBS project

For a facilitated access to the multi-instrumental data collected at observatories, the ReOBS project (see Table C1) (Chiriaco et al., 2018) aims at gathering and synchronizing multiple data sets from a given observatory in one single netcdf file at a 1 h time resolution. Chiriaco et al. (2018) more specifically deal with the data of the SIRTA observatory (Instrumented Site for Remote Sensing Atmospheric Research), from which the project has started. Since then, and in the context of ACTRIS-Fr and its aim at harmonizing the data and their access from the national AERIS data center, the ReOBS project has been extended to other observatories, P2OA among them. This is an ongoing work at P2OA, to make a single file of a large set of relevant data at 1-h time resolution, for an easier use by external users like modelists, air quality services, etc. ReOBS also merges native resolution data, for principal investigators or more specific use. It includes an additional data quality control and statistics.

## 5 Meteorological characterization of P2OA area

Here we present a meteorological characterization of P2OA based on the meteorological measurements performed over a 8-year period at P2OA, from 2015 to 2022. A flow regimes study has been previously carried out by Gueffier (2023), over the period 2015-2019, based on hierarchical ascendant classification, which gives support to our discussions below. We used here the same data set (and same period) of Gueffier (2023), for the benefit of a good data coverage for all variables, synchronization at a common hourly time base, and careful data quality control. Starting in 2015 enables us to work with the most homogeneous dataset (there are UHF data from 2010 to 2015 which still need to be processed in an homogeneous way). We first address the flow regimes, based on the radar wind profiler data and the 60 m tower data. Then we illustrate the seasonal variability of thermodynamic variables, radiation, cloud cover, precipitation and convective boundary layer depth. We end this section with the seasonal and diurnal variability of the atmospheric composition observed at PDM.

### 5.1 Flow regimes

The meteorological conditions observed at P2OA are governed by the presence of the Pyrenees' chain. Dynamically, and at the regional scale:

– the typical westerly or north-westerly synoptic winds are channeled along the chain;

– northerlies, often associated with anticyclonic conditions, can be blocked by the chain on the French side;

– southerly or south-westerly winds, often associated with a low located over the Atlantic, lead to mountain waves, flow splitting and foehn situations;

– easterlies, although rare, are also channeled along the chain. They are sometimes linked with a low situated over the Mediterranean sea, and with enhanced precipitations over the eastern part of the Pyrenees.

Figure 2 presents the wind roses found over the period 2015-2022, at three different altitude levels in the lower troposphere, based on three different instruments operated at CRA (the VHF and UHF wind profilers, and the 60-m tower). The highest level shown, at 4000 m a. s. l. shows the predominance of synoptic winds in a large southwest to northwest quadrant. At 750 m a. g. l., the wind rose looks totally different, and shows the channelling of the flow along the chain. Close to the surface (15 m above the ground), a superposition of different flow regimes is visible: in addition to the dominant westerlies, thermally-induced winds are frequent, characterized by southerly drainage (katabatic) flow from the mountain during the night, and northerly upmountain (anabatic) breeze during the day (Hulin et al., 2019). Note that the surface wind at CRA is generally weak, except in case of rare strong westerly fronts or souhterly downslope wind storms. The area, centrally close to the foothills of the Pyrenees, and protected by them, is one of the areas in France with the weakest wind.

The plain-mountain thermal wind is connected to the valley winds of the Aure Valley mentioned before. It usually sets up in calm synoptic conditions, but still can establish under significative synoptic forcing, especially when the northern Pyrenean foreland is sheltered by the mountains from southerly to southwesterly wind blowing in altitude. Hulin et al. (2019), based on the 2006-2015 dataset, found that almost 30% of days with surface breeze at CRA occur in conditions with a southerly wind component of more than 5 m s$^{-1}$ aloft. They also found an overall occurrence of 27.5% for days with surface breeze at CRA. Román-Cascón et al. (2019) have shown, for the year of 2017, that the katabatic winds at CRA lay in the sector 110-200°and have a large occurrence of more than 30% of time. Anabatic winds are less frequent, about 15%, and lay in the sector 300-50°.

Strong synoptic forcing (windspeed at 4000 m larger than 30 m s$^{-1}$), mostly corresponds to flows in the west- to northwest-erly sector in altitude. Below the altitude of the Pyrenean crest and down to the surface, they are usually channeled toward westerlies.

The foehn phenomenon, typical of the lee of mountains (Elvidge and Renfrew, 2016), is very well observed from CRA site. It is generated by synoptic southwesterly flows, generally associated with the approach of a front and with a low situated off the Bay of Biscay over the Atlantic. When the wind upstream of the mountain is strong enough, the foehn wind on the lee side can penetrate down to the surface (Scorer, 1955; Smith, 1985; Ólafsson and Bougeault, 1997). Oscillation of the whole troposphere is typically observed at CRA in southwesterly synoptic forcing (Gueffier et al., 2024), generated by mountain lee waves (Bougeault et al., 1990; Ólafsson and Bougeault, 1997)), with an occurrence of about 10% (Gueffier et al., 2024). Note that the large occurrence of southerly winds in the windrose of Fig. 2c includes the katabatic downslope winds from the Aure Valley, which is predominant at that height.

## 5.2 Seasonal variability

Figure 3 presents the seasonal and inter-annual variability of the monthly mean diurnal cycles of air temperature and water vapour mixing ratio (both at 2 m a. g. l.) at CRA and at PDM, and of downward shortwave radiation (at the top of the 60 m tower) at CRA. The monthly composite days are shown separately for each year of the 2015-2022 period, which also highlights the inter-annual variability for a given month.

The downward shortwave radiation seasonal variability is typical of the mid-latitude. June-July-August are the hottest and most humid months, associated with the highest shortwave downward radiation. The diurnal cycle of temperature at CRA is

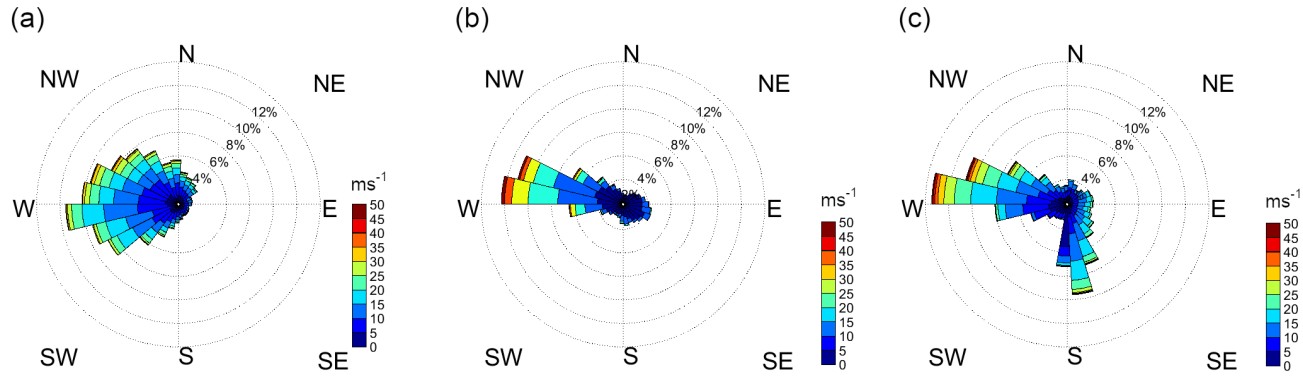

**Figure 2.** Wind roses at (a) 3975 m a. s. l, (b) 750 m a. g. l and (c) 15 m a. g. l at CRA over the period 2015-2022.

marked in any month, even if of lesser amplitude in January and February. It is noticeable to see how it can remain high in September at the end of summer, and be significant even in December on some specific warmer years. Those months often show anticyclonic dry and calm conditions, with clear sky and marked diurnal cycle. The diurnal cycle of moisture at CRA
is less marked than that of temperature, partly because the composite is disturbed by non-diurnal variability like fronts or mesoscale advections.

It is interesting to notice the different behaviour of the diurnal cycles at PDM relatively to CRA. From April to October, water vapour mixing ratio (temperature) shows a larger (smaller) diurnal amplitude at PDM than at CRA. This is likely due to the possible occurrence of deep CBL at midday during this period, that may overwhelm PDM top during daytime but leaves
it in the free troposphere during the night. On the contrary, in winter months, PDM summit is mainly in the free troposphere, with less marked diurnal cycle than the rest of the year, and than that observed at CRA.

The inter-annual variability is usually large, often of the same order of magnitude as the diurnal variability (December and February are good examples of large inter-annual variability here), except in summer here (it is very small in August for this period, but might be larger for another period).

Figure 4 shows the seasonal variability of the cloud cover (monthly averages), based on the RAPACE total sky imager and the ELIFAN algorithm (Lothon et al., 2019). The monthly standard deviation is expectedly large (not shown), about 40%, larger than the seasonal variability (less than 10%). Still, April and May have on average larger cloud cover than September and October. Considering monthly averages calculated in the morning (bewteen 3 and 4 h after sunrise) or in the afternoon (between 4 h and 3 h before sunset) reveals a slight diurnal cycle, explained by a significant number of days with afternoon convection.
Interestingly, this diurnal cycle vanishes in summer. This could be explained by the occurrence in summer of nighttime stratus clouds which dissipate in the late morning or afternoon, compensating days with clear mornings but afternoon convection. Those stratus clouds can notably come from blocked northerly moist flows, or from moisture coming from the daytime moist convection over the mountain which is concentrated at the bottom of the valleys and the plain during the night. Summer has a large occurence of moist low layers in the northern Pyrenees, associated with a rather vegetated and moist surface, relatively

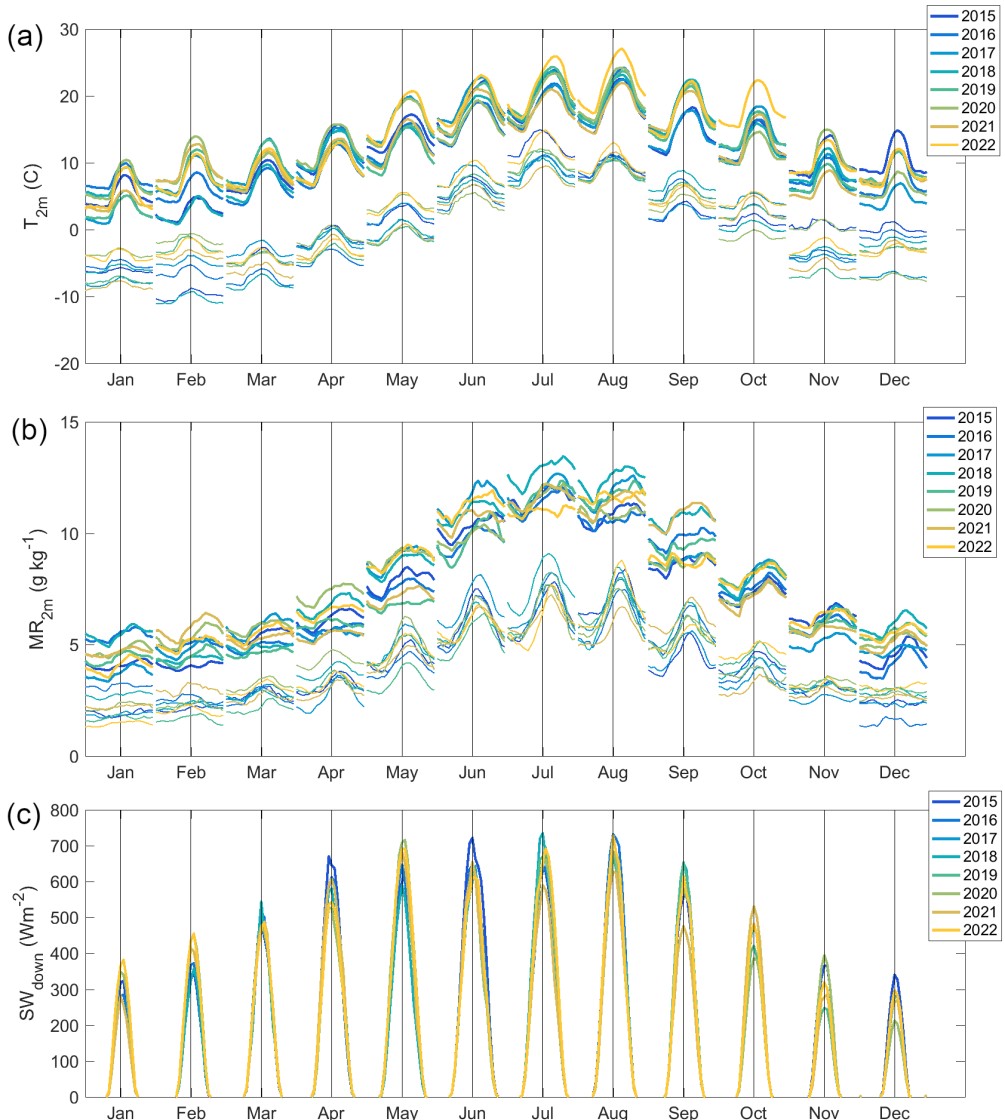

**Figure 3.** Seasonal and inter-annual variability of monthly-averaged diurnal cycles of (a) 2-m temperature at (thick line) CRA and (thin line) PDM, (b) 2-m water vapour mixing ratio at (thick line) CRA and (thin line) PDM, (c) downward short wave radiation at CRA, on top of the 60 m tower.

to the southern Spanish side. The diurnal cycle also disappears in winter, with the decreased shortwave radiation, drier air and suppressed convection.

In Figure 5, the precipitations for the period 2015-2022 are shown, through the mean monthly cumulative rain, the monthly mean rain rates for precipitations larger than 3 mm h$^{-1}$, and the monthly maximum rainrate encountered. The use of those three variables enables to distinguish between frequent low precipitation events and convective intense rain events. November,

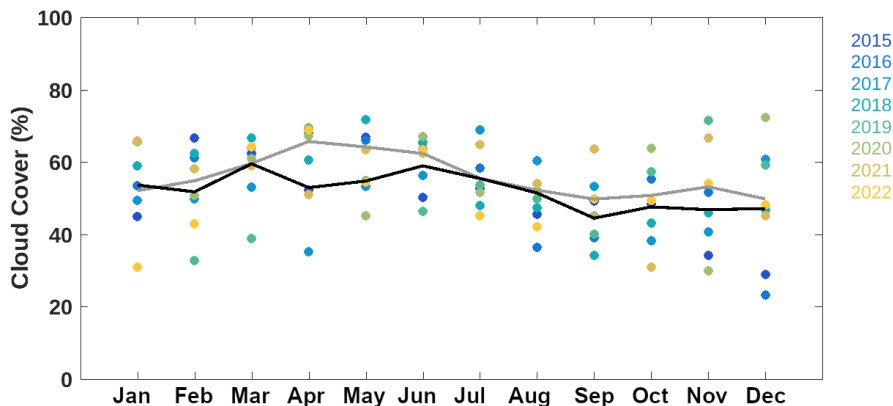

**Figure 4.** Seasonal and inter-annual variability of monthly-averaged cloud cover, for the period 2012-2019. Each dot is a monthly average. The black line is the averaged for morning hours (3h after sunrise), and the grey line for afternoon hours (3h before sunset).

January and February have the largest cumulative rain, associated with largest number of rainy days. May, June, July, August are the month with the highest maximum rainrates encountered, associated with convective storms. May and June have the largest mean rainrates, when considering only significant rain events with precipitation rates larger than 3 mm h$^{-1}$. These months combine the chance of storms and the occurrence of moist blocked northerly flows. The driest months are September and October.

Figure 6 shows the convective boundary layer (CBL) depth $Z_i$, estimated from the UHF wind profiler with the algorithm developed by Philibert et al. (2024), based on the detection of a local maximum of the air refractive index structure coefficient and a minimum of vertical velocity standard deviation. The algorithm also takes temporal continuity into account. Note that only convective boundary layers are considered here, which implies that only cases with no rain and no fog allow for a CBL depth estimation. As expected, the cold winter months of November, December and January have significantly smaller CBL

depth relatively to other months. Spring and summer interestingly reach similar CBL depth, despite the variation of incident shortwave radiation from one season to the other. This can be explained by several features. In summer in the mountain area, the heating of the valley atmospheres is stronger than above flat terrain, mainly due to reduced volume (concept of topographic amplification, see Steinacker, 1984; Whiteman, 1990), and generates enhanced convection due to increased instability. Cumulus and congestus clouds develop actively all along the mountain ridge during fair weather days, inducing a mesoscale subsidence

near the mountain base (De Wekker, 2008). Therefore, the CBL at CRA is often capped by a significant subsidence (Pietersen et al., 2015; Blay-Carreras et al., 2014b) in this season, which limits the CBL growth. In April and May convection is less active over the mountain, while the sensible heat flux can be as large as in summer or even larger, because solar heating is significant while air masses are still cold (typically during post-frontal situations). This favors CBL growth. The month of September experiences similar CBL depth (and sensible heat fluxes) than April and May, in clearer sky, less rain, compensated

by a smaller temperature gradient close to surface. February and October consistently show intermediate CBL depths, between

the two groups winter/summer. Standard deviation of $Z_i$ within one month ranges from 100 to 500 m, with larger values in summer and at midday. It is around 300 m on average.

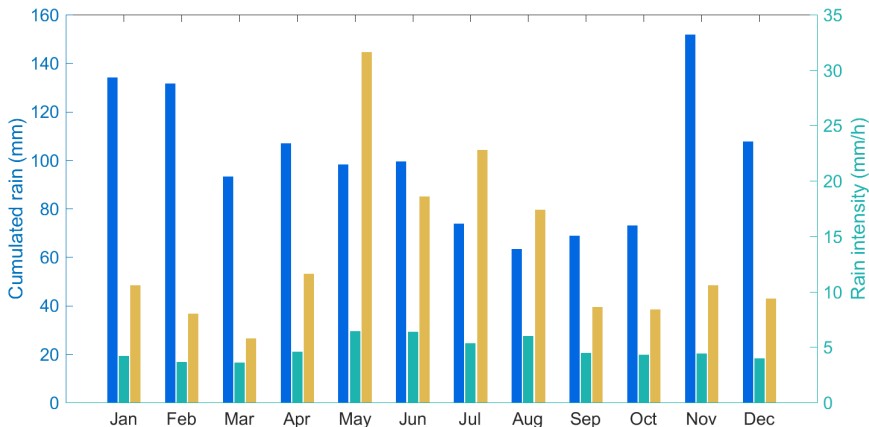

**Figure 5.** Monthly cumulative rain amounts (blue, left y-axis), monthly mean rainrates for precipitations > 3 mm (green, right y-axis), and monthly maximum encountered rainrates (brown, right y-axis) for the 2015-2022 period, at CRA.

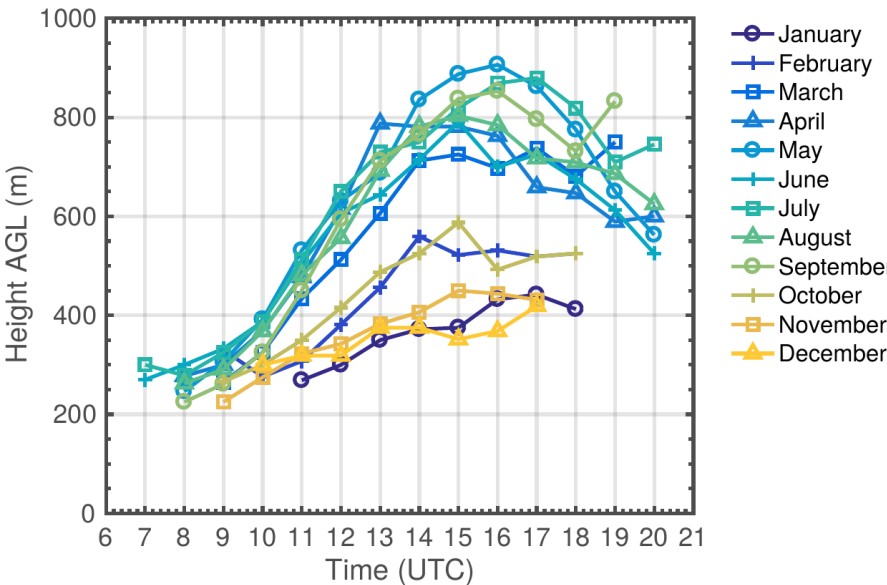

**Figure 6.** Composite diurnal variation of CBL depth $Z_i$ at CRA, averaged over 2015-2022, for each month of the year. $Z_i$ estimates shown here were retrieved by use of CALOTRITON algorithm (Philibert et al., 2024).

## 5.3 Long-term temperature trend

The unique PDM historical temperature time series built since the 1880's by the Observatoire Midi-Pyrénées and Météo-France were first studied by Bücher and Dessens (1991) and Dessens and Bücher (1995). Bücher and Dessens (1991) estimated to +0.83°C the temperature trend from 1882 to 1970. Dessens and Bücher (1995) revealed that over the 100-year period [1882-1984], the daily maximum temperature slightly decreased of 0.5°C/100 yr, while the daily minimum (nightime) temperature significantly increased of 2.59°C/100 yr. This means that the amplitude of the diurnal cycle decreased of about 2.9°C/100 yr, which is very significant. They showed that this temperature trend was associated with an increase in both the relative humidity and cloud cover. Figure 7 shows an update of this long-term time series until 2020, along with the temperature measured at 2 m at CRA since 1991. Recent data since 1984 used here come from the Météo-France synoptic stations at both sites.

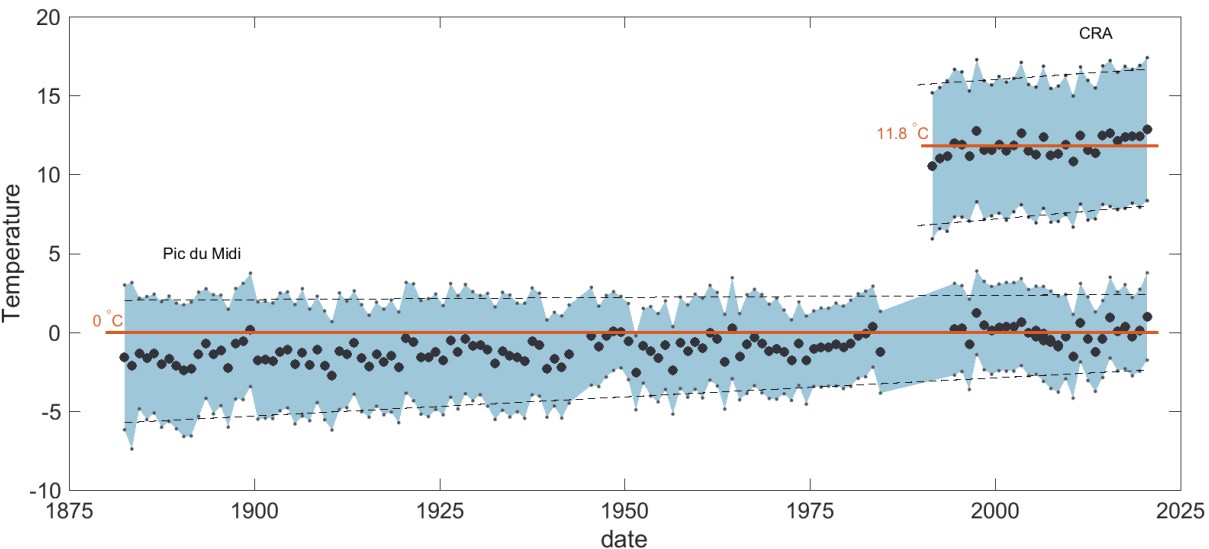

**Figure 7.** Long-term series of the annual mean of daily minimum and maximum temperatures at PDM and CRA (small dots and envelop). Bold dots represent $\frac{1}{2}(T_{max} + T_{min})$. The level of 0°C is indicated for the historical series of PDM, as indicative of the retreat of glaciers at this altitude in the Pyrenees. The level of 11.8°C is indicated as temperature reference for CRA, which is the averaged temperature over the period [1990-2000]. For each minimum or maximum temperature time series, linear regression is indicated with dashed lines. Slopes are 2.4°C/100 yr, 0.3°C/100 yr, 1.2°C/30 yr, and 0.9°C/30 yr for respectively the PDM minimum and maximum temperature and the CRA minimum and maximum temperature.

Temperatures shown here are the annual mean of the daily minimum and maximum temperatures $T_{min}$ and $T_{max}$ respectively, as well as the half-sum $\frac{1}{2}(T_{max} + T_{min})$, in order to remain homogeneous with the series studied by Dessens and Bücher (1995) and consistent with the international convention. This half-sum is conventionally used as representative of the yearly-averaged temperature.

At CRA, we can observe that the last seven years (2014-2020) were all above the current temperature reference for this site, taken as the average over the period [1990-2000] (11.8°C). A linear regression over the annual mean temperature series gives an increase of temperature of 1.1±0.4°C over the 30 years of measurements. But the regression coefficient is only 0.26, and the period is not long enough to give a robust trend estimate.

At PDM it is first important to notice that the mean temperature at PDM has exceeded zero in the 1980's. This has a strong impact on Pyrenean glaciers (Marti et al., 2015). A new linear trend estimate on this series would give a trend of +1.3±0.1°C/100 yr when calculated over the total period, +0.9±0.2°C/100 yr for the first 100 years, and +1.5±0.2°C/100 yr for the last 100 years. This is a significant difference that may reveal an acceleration of warming in the last three decades. Caution needs to be taken though, since a period of 10 years is missing in the series from 1984 to 1994.

Note that compilation of historical ozone measurements has shown an increase of tropospheric ozone by a factor 5 from the 1900's to the early 1990's (Marenco et al., 1994) but a stabilization around 45 nmol mol$^{-1}$ since then and until the 2000's (Chevalier et al., 2007).

## 5.4 Atmospheric composition

Gases and particles measurements performed at PDM are to some extent representative of the air composition in the free
troposphere (see discussion in Section 5.4). These measurements include reactive and greenhouse gases, as well as physical and chemical properties of suspended particles. In situ gas measurements are also available at the lowland site CRA (Table 1). As illustration, Figure 8 shows diurnal composites of ozone (ACTRIS-Fr), carbon dioxide (ICOS), and methane mole fractions (ICOS), and total suspended particle concentrations, separated on a seasonal basis. A five-year period has been chosen to get averaged values with a good multi-annual representativity, and optimized data coverage: January 2015 to December 2019 for
PDM (consistent with the work by Gueffier (2023) and Gueffier et al. (2024)) ; May 2019 to April 2024 for $CO_2$ and CH4 at CRA (as measurement started more recently there, in April 2019) [9].

Figures 8a (PDM) and 8b (CRA) commonly show that the ozone concentration is larger in spring and summer, in link with increased photochemical activity in the troposphere during those seasons (e.g. Chevalier et al., 2007). Further, the concentrations range between higher values in the free troposphere (Fig. 8a) than in the continental atmospheric boundary layer (Fig. 8b),
as also evidenced for western Europe by Chevalier et al. (2007). At both sites, a diurnal cycle is well visible in summer, but of lesser amplitude in spring and fall, and almost absent in winter. The cycle at PDM (Fig. 8a) shows an ozone minimum short after noon but maximum values during the night, which is mainly due to the influence of anabatic transport from the valleys and the plain to the mountain summits. This had firstly been identified at PDM by Marenco (1986) based on a 1-year measurement campaign in 1982. This author raised the idea that air sampled at PDM during the warm season was a mix of
boundary layer and free-tropospheric air, with proportions varying along the day time – free tropospheric conditions being experienced during the night. The anabatic transport thus results in a decrease of ozone concentration during the daytime, as the air from the lower levels is poorer in ozone relative to the free troposphere at the continental scale, as evidenced above.

---

[9]Due to the rapid multi-annual anthropogenic trend of $CO_2$ and CH4, the absolute values presented here for PDM and CRA at different periods should not be compared.

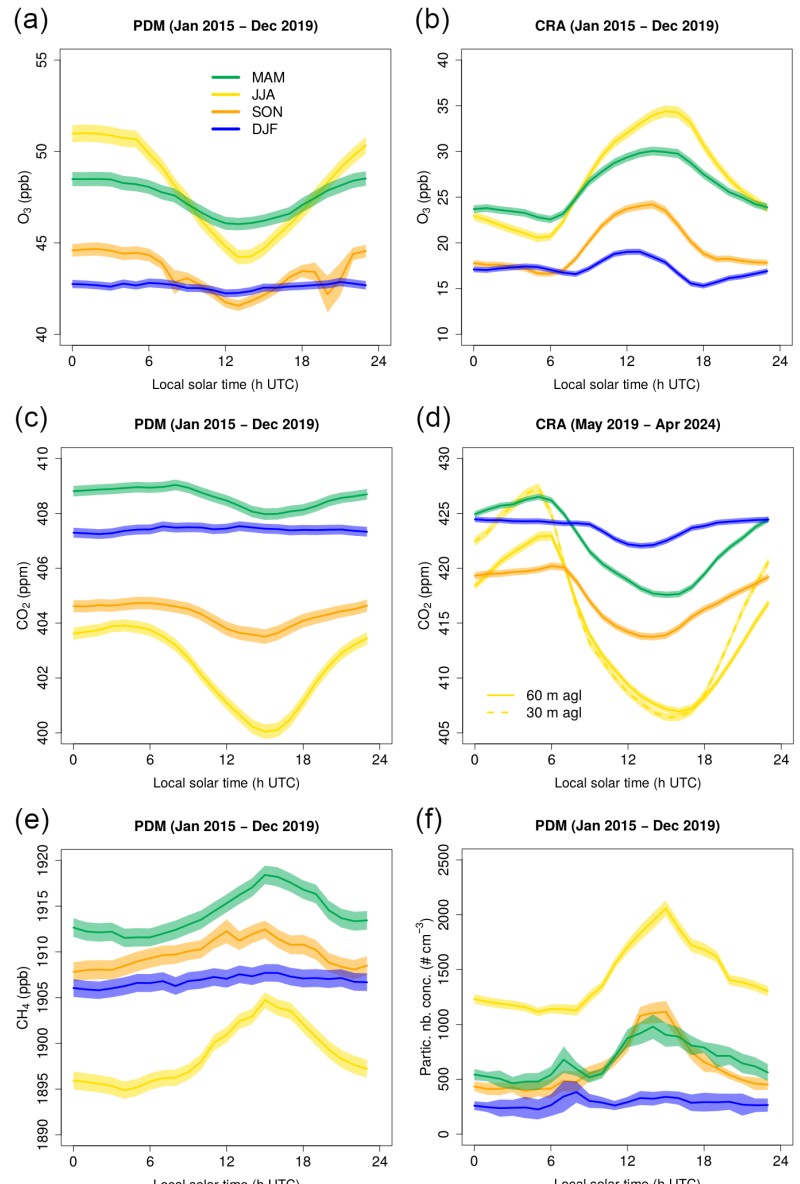

**Figure 8.** Composite diurnal cycle by season of mole fractions of (a-b) ozone, (c-d) carbon dioxide, (e) methane, and (f) total suspended particle number concentration observed at PDM (a,c,e,f) and at CRA (b,d): DJF = December-January-February; MAM = March-April-May; JJA = June-July-August; SON = September-October-November. In panel (d) all curves represent air taken at 60 m agl, except the dashed JJA curve representing air taken at 30 m agl. The uncertainty ranges correspond to plus or minus the standard error of the sample. Mean values are considered in (a-e) but median values were preferred in (f), due to the distribution skewness for particles.

The anabatic ozone decrease is stronger during the hot seasons where anabatic transport most occurs, and is indeed typical of mountain summit observatories (Tsamalis et al., 2014). Concerning the flat land site CRA (Fig. 8b), the ozone diurnal evolu-
540 tion is typical of the cycle most often observed in the continental boundary layer, with a maximum in late afternoon, and a minimum at sunrise. During the night, the ozone decay in the stable boundary layer is due to ozone surface deposition (and possibly to ozone titration by NO, but NO concentrations are expected to be low in this rural area). As soon as the sun rises, entrainment of higher concentrations of ozone from aloft within the growing convective boundary layer contributes to make ozone concentration grow near the surface (as illustrated in Tsamalis et al. (2014)). Photoproduction can also contribute to
545 the daytime ozone build-up, but further investigation is needed to assess this contribution with consideration of local NOx measurements.

The influence of anabatic transport was studied specifically for PDM by Gheusi et al. (2011); Tsamalis et al. (2014); Hulin et al. (2019). These studies further illustrate that it affects other atmospheric species, as soon as a vertical concentration gradient exists in the regional background atmospheric profile. This is the case for carbon dioxide ($CO_2$), which undergoes a similar
influence (Fig. 8c): during the daytime, the anabatic upflows transport air depleted in $CO_2$ relative to free tropospheric air at the height of PDM, due to increased photosynthetic activity near the surface in the valleys and the plain. This again results in a marked diurnal cycle of $CO_2$ during summer, of weak amplitude in spring and fall, and almost no diurnal variability during winter. Such influence is also observed at other mountain-top observatories in the world (e.g. Necki et al., 2003). At the seasonal scale, the tropospheric background in $CO_2$ decreases as the photosynthetic activity increases during the vegetation season, so
concentrations are maximum in winter and minimum in summer. Note that a multi-annual increase of about 2 ppm per year was estimated for $CO_2$ at PDM over the period 2015-2019 by Gueffier et al. (2024), consistent with the global anthropogenic trend observed worldwide (Friedlingstein et al., 2023). $CO_2$ measurement at CRA (Fig. 8d) shows similar seasonal trends as at PDM concerning the absolute $CO_2$ levels. Nevertheless, the amplitude of the diurnal cycle is wider, especially in summer. Nighttime $CO_2$ build-up is observed until sunrise, presumably due to soil respiration and $CO_2$ accumulation near the ground in the stable
boudary-layer. This idea is supported by stronger $CO_2$ concentrations found at 30 m than at 60 m above the ground during the night. After sunrise, mixing within the developing convective boundary layer dilutes the $CO_2$ previously accumulated near the ground, and the $CO_2$ curves at 30 m and 60 m tend to overlap during the day.

In the P2OA rural and moutainous region, methane ($CH_4$) mainly comes from agricultural activity. However, oxydition of methane by OH radicals results in a seasonal decrease of $CH_4$ when oxydation is most important (Necki et al., 2003). This
explains the maximum of methane concentration in winter, and the minimum in summer (Fig. 8e). Similarly to other mountain observatories (Necki et al., 2003), $CH_4$ at PDM displays a marked diurnal cycle in summer with a maximum in the daytime, here again related to transport by anabatic flows, conveying air richer in $CH_4$ from the low lands. This diurnal cycle is again less pronounced in spring and fall, and absent in winter, consistent with the seasonal varibility of the occurrence and intensity of thermally-driven circulations at PDM (Hulin et al., 2019).
As at high Alpine sites, anabatic transport at PDM (Fig. 8f) is one of the most decisive factors contributing to aerosol concentration variability (Collaud Coen et al., 2011; Herrmann et al., 2015). Still for comparison, Sun et al. (2021) showed also

that at Zugspitze-Schneefernerhaus (2671 m) and Jungfraujoch (3580 m) higher concentration and stronger diurnal variability were observed in the warm season, while lower concentration and less distinct diurnal variability in the cold season.

## 6 Illustrative studies based on P2OA

In this section, we present examples of various applications of P2OA dataset and experimental sites, concerning:

- atmospheric process studies based on the long-term series or on hosted field experiments (Section 6.1);

- instrumentation test campaigns (Section 6.2);

- evaluation of numerical weather prediction models (Section 6.3).

All field experiments hosted at P2OA are listed online[10], with indications of the hosting site (PDM, CRA, both), addressed scientific topic, and involved research laboratories. Here in the following subsections, we focus on a few chosen projects or studies illustrative of the scientific potentials of P2OA.

### 6.1 Process studies

### 6.1.1 Surface/atmosphere interaction

The instrumentation of P2OA, and in particular at CRA, is especially appropriate for the study of the atmospheric boundary layer dynamics and surface/atmosphere interaction. In 2011, an international field experiment was hosted at P2OA for the study of the transition from daytime convective boundary layer to the stable nocturnal boundary layer: the Boundary Layer Late Afternoon and Sunset Turbulence (BLLAST: Lothon et al., 2014, https://bllast.aeris-data.fr). This transitional phase of the diurnal cycle was the unique focus of this project and field campaign. During 3 weeks, research groups from Europe and the USA have gathered a large number of complementary devices in order to densely observe the atmospheric boundary layer from surface to the top, and from midday to the night. Radiosoundings, tethered balloons, manned aircrafts, unmanned airplanes were operated during 12 favorable days, and continuous measurements were with instrumented towers deployed over various vegetation covers, and with remote sensing devices (lidar, sodar, radiometer, ceilometer, camera), which all were added to the existing permanent instrumentation. From the collected dataset, a fine description of the turbulence decay was done (e. g. Lothon et al., 2014; Darbieu et al., 2015; Nilsson et al., 2016a, b), which revealed how the decay remains normalized and faithful to the Deardoff model quasi-stationarilly until the surface flux gets really too small to maintain the mixing. Starting then, the turbulence decay gets faster, the turbulence structure changes, with larger turbulence scales in the boundary layer and smaller in the surface layer. The shape of the energy spectra also changes. It was also shown that a pre-residual layer can be defined before the residual layer overlying the stable nocturnal layer. This pre-residual layer corresponds to a layer between the top inversion and the top of the turbulence layer coupled to the surface. At that time, again, the surface fluxes are not strong enough to maintain the mixing up to the top inversion and previous daytime convective boundary layer top. The scheme in

---

[10]https://p2oa.aeris-data.fr/field-campaigns/

Fig. 9 summaries those findings. In this figure, time is represented with normalized dimensionless time $\hat{t}$, which is typical in turbulence decay studies(Nadeau et al., 2011). $\hat{t}$ is based on the period from maximum to null buoyancy flux:

$$\hat{t} = \frac{t - t_{max}}{t_{zero} - t_{max}}, \tag{1}$$

where $t_{max}$ is the time during midday when surface buoyancy flux is maximum, and $t_{zero}$ is the time when surface buoyancy flux gets to zero later in the day. Thus, $\hat{t}$ equal zero at maxium buoyancy and 1 at zero buoyancy flux. Several other features were finely studied, revealing for example the importance of the advection of small scale heterogeneities (Cuxart et al., 2016), the mystery of the Lifting Temperature Minimum a few tens of centimeters above surface during the stabilization process (Blay-Carreras et al., 2015), the difficulty of scaling the turbulence decay (Nilsson et al., 2016a; El Guernaoui et al., 2016), the occurrence of counter-gradient heat fluxes (Blay-Carreras et al., 2014a), the uncertainty of Monin-Obukhov Similarity Theory (MOST) (Kooijmans and Hartogensis, 2016), or the interactions between the drainage flows, gravity waves and turbulence (Román-Cascón et al., 2015).

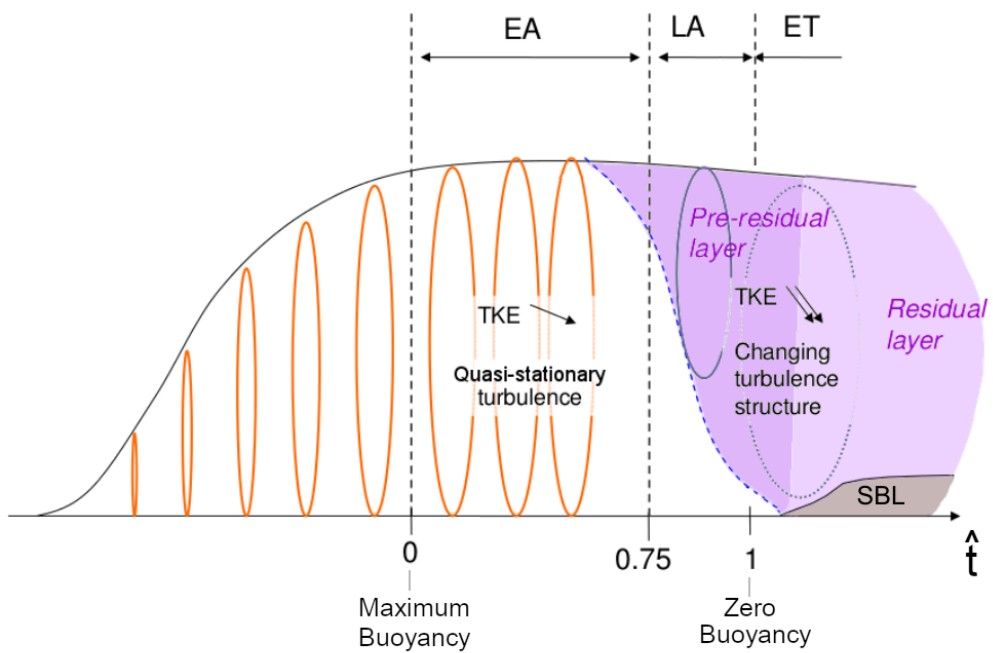

**Figure 9.** A scheme of the atmospheric boundary layer diurnal cycle and turbulence structure evolution, based on the results of the BLLAST project (Lothon et al., 2014). $\hat{t}$ is normalized dimensionless time. EA: Early Afternoon, LA: Late Afternoon and ET: Evening Transition are defined from there. TKE: Turbulente Kinetic Energy, SBL: Stable Boundary Layer. Ellipses represent the structure, size and coupling of vertical velocity eddies.

BLLAST only slightly addressed the role of surface heterogeneity, while the latter raised several questions, not only during the afternoon transition, but more generally. In 2023, P2OA hosted a new field experiment focused on the impact of surface

heterogeneity on the atmosphere: the MOSAI project (Model and Observation Surface-Atmosphere Interactions, https://mosai.aeris-data.fr/, Lohou et al. (2023)). MOSAI addresses three questions :

– What is the representativeness of the 60 m tower flux measurements or other permanent measurements with respect to the surrounding heterogeneous landscapes (prairies, forests, crops, and small villages)?

– How can we develop appropriate methodologies to evaluate the NWP and Climate models on surface/atmosphere interactions?

– Can we improve the representation of surface-atmosphere coupling in NWP and Climate models?

To address those questions, three one-year field experiments were planned, at three instrumented sites in France: Meteopole in 2021 (Canut et al., 2019), SIRTA in 2022 (Haeffelin et al., 2005), and P2OA in 2023. Each time, the 4 or 5 most representative vegetation covers of a 5 km × 5 km model mesh around the permanent tower are instrumented with surface flux towers. In 2023 at P2OA, three 15-day intensive observation periods (in April, August and December) were added for the study of a transition forest-to-culture, with a tethered balloon, remotely piloted airplane systems (RPAS) and several towers instrumented at different levels above and within the forest. This should enable us to finely study the vertical structure of the local transition, with different sublayers (internal boundary layer, equilibrium boundary layer, etc., Bou-Zeid et al., 2020).

### 6.1.2 Atmospheric composition

– **Impact of meteorology on atmospheric composition at PDM**

It has been long recognized (e.g. Keeling et al., 1976) that atmospheric composition measurements conducted on top of high mountains are better representative of the troposphere at the global scale than continental low altitude stations, which undergo local influences. Comparing ozone mole fractions measured at PDM and other mountain stations in Europe on the one hand, versus airborne measurements in the free troposphere on the other hand, Chevalier et al. (2007) indeed found agreement within 8% with the reference airborne profiles for stations above 2000 m a.s.l., provided multi-year averages were considered.

Summit observatories may nevertheless be influenced, at least part of the time, by local or regional emissions transported by boundary layer processes. Two field campaigns – 'Pic 2005 and 'Pic 2010' – were designed to address this question for PDM. Pic 2005 (Gheusi et al., 2011) revealed that during summer fair-weather days, ozone measurements in the daytime at PDM were representative of the mixing of layers present between 1000 and 2000 m a.s.l. above the Pyrenean foreland (CRA). By means of a Lagrangian transport model integrating ozone photochemistry, Tsamalis et al. (2014) showed, for two fair-weather days during Pic 2005, that best adjustements to observed ozone diurnal variations at PDM were obtained when 14 to 57% (depending on the day) of air from the boundary layer was incorporated into free-tropospheric air before reaching PDM.

During Pic 2010, meteoroglogical and ozone radiosoundings were launched simultaneously from CRA and from a valley bottom very close to PDM (Hulin et al., 2019). These profiles allowed the characterization of a humid venting layer

formed during the daytime above the Pyrenees by anabatic pumping of low-level air, then exported above the plain by the altitude return branch of the plain-to-mountain breeze below 2000 m. This is here well illustrated by a numerical simulation of an anabatic day (Fig. 10). In such conditions, PDM appears to be influenced during the afternoon by local weak southerly wind (which is typically observed) conveying the Pyrenean venting layer towards the plain.

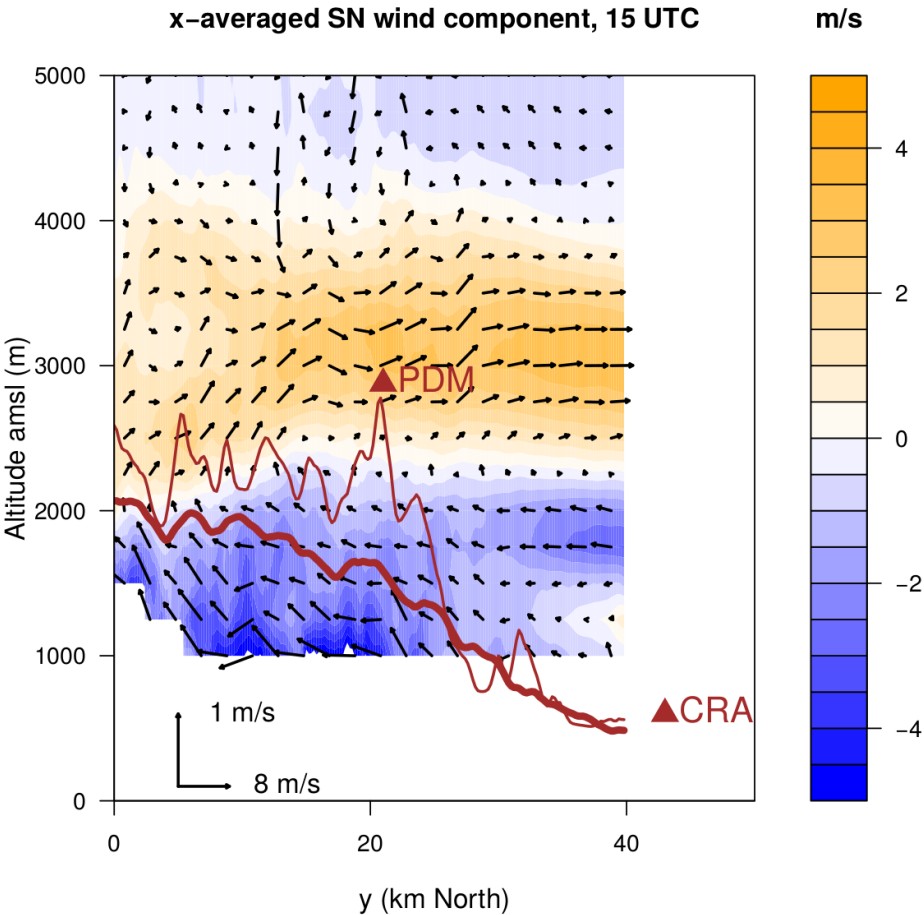

**Figure 10.** High-resolution (200 m) simulation with Meso-NH (Lac et al., 2018) of a typical summer sunny afternoon with a well-developed plain-mountain thermal circulation (10 July 2010, 1500 UTC; PDM being located at the center of a $40 \times 40$ km$^2$ domain). This south-north vertical cross-section ($yz$ plane) shows model fields averaged over 40 km along the zonal ($x$) direction. The vectors show the projection of the ($x$-averaged) wind on the section plane, while the color code emphasizes the $y$ component. The brown thin line shows the terrain profile at the longitude of PDM, while the bold line represents the $x$-averaged terrain elevation in the model domain. PDM (represented at its real altitude) peaks markedly above the $x$-averaged terrain altitude, but only 100 m above the model terrain at this place.

To put the above studies in their climatological context, Hulin et al. (2019) also explored the occurrence of thermally-driven circulations at P2OA over a 10-year period (2006-2015), and their impact on air composition at PDM. Detection methods of thermal circulations in P2OA area allowed to separate ensembles of days with or without anabatic influ-

ence at PDM, revealing contrasted diurnal evolutions of the observed atmospheric species (consistent with the diurnal composites shown in Fig. 8).

Gueffier et al. (2024) went beyond the specific influence of thermally-driven circulations, and explored more generally the influence of synoptic meteorology on air composition at PDM. Considering 5-year of meteorological data collected at P2OA (2015-2019), weather regimes were distinguished by hierarchical clustering. The most characteristic ensembles that emerged were: fair calm weather days (favorable to thermally-driven circulations); disturbed weather with westerly advection; and south foehn conditions. Marked differences were found between the meteorological clusters when considering the air contents in Rn, $O_3$, CO, $CH_4$, $CO_2$ and particles. Among other results, it was shown that (i) air driven to PDM by south foehn had mostly a free-tropospheric signature; (ii) despite enhanced anabatic influence, the regional free tropospheric influence remains dominant during anticyclonic fair weather days; (iii) disturbed weather caused intense mixing of the lower troposphere at the regional scale, and thus the influence of regional emission sources is stronger.

– **Aerosol properties**

Guillaume et al. (2008) succeeded in simulating with a global model (ORISAM-TM4, Guillaume et al., 2007) the temporal evolution of black carbon (BC) and total organic carbon (OC) in the aerosol measured at PDM from weekly filter samples. They thus showed that large-scale BC pollution is most prominent at this site compared to possible local influences, especially during heat waves as the major one during the 2003 summer. In addition, formation of secondary organic aerosols was found to be a major component of OC in such meteoroglogical conditions.

As an indubitable evidence of hemispheric transport to PDM, radioactive fallout due to explosions of three reactors of the nuclear power plant in Fukushima (Japan) on 12-15 March 2011, was detected at PDM in an aerosol filter sample (Evrard et al., 2012). The presence of 131-iodine in aerosols ($200\pm6$ $\mu$Bq m$^{-3}$) indicated that the radioactive cloud reached France between 22 and 29 March, i.e. less than two weeks after the initial emissions, as the IRSN measurements devices also indicated.

More recently, Tinorua et al. (2023) performed a study based on two-years measurements (2019-2020) of BC microphysical and optical properties at PDM, using specific instruments to complement the ACTRIS instrumental set. They showed that among the worldwide existing long-term monitoring sites, PDM experiences only occasional influence of the planetary boundary layer, making it an ideal site for characterizing free tropospheric BC. Moreover, their classification of the dominant aerosol type using the spectral aerosol optical properties indicated that BC was the predominant absorption component of aerosols at PDM, and controlled the variation of Single Scattering Albedo throughout the two years.

– **Greenhouse gases**

P2OA is part of the national ICOS-France network, but PDM and CRA not yet labelled as European ICOS-ERIC stations. They nevertheless both contribute to the European datasets compiled annually by ICOS for $CO_2$ (Bergamaschi et al., 2024a) and $CH_4$ (Bergamaschi et al., 2024b) In particular, PDM station was at the heart of the evaluation of the method

for spike detection algorithm which is now applied to all the stations in the ICOS network (El Yazidi et al., 2018; Tenkanen et al., 2021; Cristofanelli et al., 2023). P2OA data have also been used to study the impact of European droughts on the $CO_2$ concentrations and fluxes (Ramonet et al., 2020; Thompson et al., 2020; He et al., 2023), as well as to study global and European methane balances (Saunois et al., 2020; Szénási et al., 2021; Thompson et al., 2021).

– **Atmospheric Mercury**

Mercury (Hg) is a heavy metal that is dispersed globally in the gas phase following its emissions from volcanoes and human activities. The atmospheric lifetime of Hg is not well constrained due to its complex redox chemistry, making model predictions of Hg deposition and ecosystem loading difficult. At P2OA, we have generated from 2010-2014 one of the longest high-altitude atmospheric Hg datasets, using automated instruments that quantified gaseous elemental Hg(0), gaseous oxidized Hg(II) and aerosol Hg(II) dynamics at a 2 h resolution (Fu et al., 2016a; Marusczak et al., 2017). Together with experimental rainfall Hg photoreduction rates (Yang et al., 2019), these data have helped to better constrain atmospheric Hg redox reactions in global Hg chemistry and transport models (Saiz-Lopez et al., 2018). We have also developed unique methods to sample and measure the stable isotope composition of atmospheric Hg(0) and Hg(II) in gas, aerosol and precipitation, which inform on deposition pathways and fluxes (Fu et al., 2016b, 2021; Enrico et al., 2016). The data are freely available through the AERIS/GMOS and iGOS4M datahubs (see Table C1).

– **Microplastics and other trace species**

P2OA also hosted punctual campaigns on the atmospheric chemistry and transport of diverse environment-impacting species such as halogens and selenium (Suess et al., 2019), formaldehyde (Prados-Roman et al., 2020) and microplastics (Allen et al., 2021). Over a period of four months in summer 2017, polymer fragments and fibers ranging in size from 3.5 to 53 micrometers were observed at relatively low (0.25 microplastics $m^{-3}$) but significant levels. The polymers identified, polyethylene, polystyrene, polyvinyl chloride, polyethylene terephthalate and polypropylene, are all known for their use in packaging. The origin of these microplastics has been studied by back-trajectory modelling of the air masses observed at PDM. Many of these trajectories have their origin in Africa, the Atlantic Ocean and North America, indicating that an intercontinental transport is at the origin of the microplastics detected at PDM.

### 6.1.3 Exploring Transient Luminous Events

In the late 1980s, a new field of research opened with the discovery of the Transient Luminous Events (TLEs), which includes now sprites, 'elves' (Emission of Light and Very low-frequency perturbations from Electromagnetic pulse Sources), jets and other electrical phenomena occurring above thunderstorms (Füllekrug et al., 2006). The first European TLE-dedicated observations were obtained at PDM, in 2000 (Neubert et al., 2001). In the following years, several European teams joined in Eurosprite during summer and fall campaigns leading to over 700 TLE images being captured in the period from 2000 to 2008 (Neubert et al., 2008). The two observation sites of P2OA, equipped with a remote-controlled sensitive camera system, contributed to a climatology of TLEs in Europe (Arnone et al., 2020) and a large number of process studies (e.g. Soula et al., 2015, 2017; Gomez Kuri et al., 2021; Tomicic et al., 2021).

During five successive nights from 16 to 21 January 2017, several storm systems over the Mediterranean Sea, highly productive in TLEs, were monitored with the camera at PDM. A total of 589 TLE events were recorded and analyzed in the thesis by Gomez Kuri (2021). This large number of TLEs and the diversity of the storm systems capable of producing TLEs allowed to support previous observations or reinforce some theories on mechanisms and conditions of their production. Figure 11 shows the continuity of the cloud-to-ground (CG) lightning flash activity observed from PDM with Météorage[11] network during five days of the winter period within the region pointed by the camera. There was an increasing flash rate during several cycles corresponding with individual cells at the beginning of the period and an evolution to larger developments of storms such as mesoscale convective systems in the second part of the period. The TLEs were observed over the five nights, with specific time intervals more favorable to some TLE categories like elves (green histogram in Fig. 11) when the cloud-to-ground flash rate was large (around 20:00 UT on 20 January) and sprites (red histogram) a few hours later while the flash rate decreased substantially. The analysis of the location of each TLE in its storm context shows that sprites were often produced over stratiform regions of mesoscale convective systems, during their dissipating stage, and associated with a positive stroke, the peak current of which was on average about 107 kA. Elves mainly occurred in systems of strong convection, during the mature stage which produced negative strokes of high peak current. Elves are more likely to occur with clouds that have large vertical but small horizontal extensions whereas sprites occur over clouds of smaller vertical but larger horizontal extensions (Gomez Kuri, 2021). In the night of 19 January, out of 106 TLEs observed above a mesoscale convective system developing between south of Ebro delta and the Balearic Islands, 97% of 29 elves occurred over sea while 45% of sprites occurred over land. It is in accordance to other observations and to larger peak current values for strokes over sea.

---

[11]https://www.meteorage.com/fr/

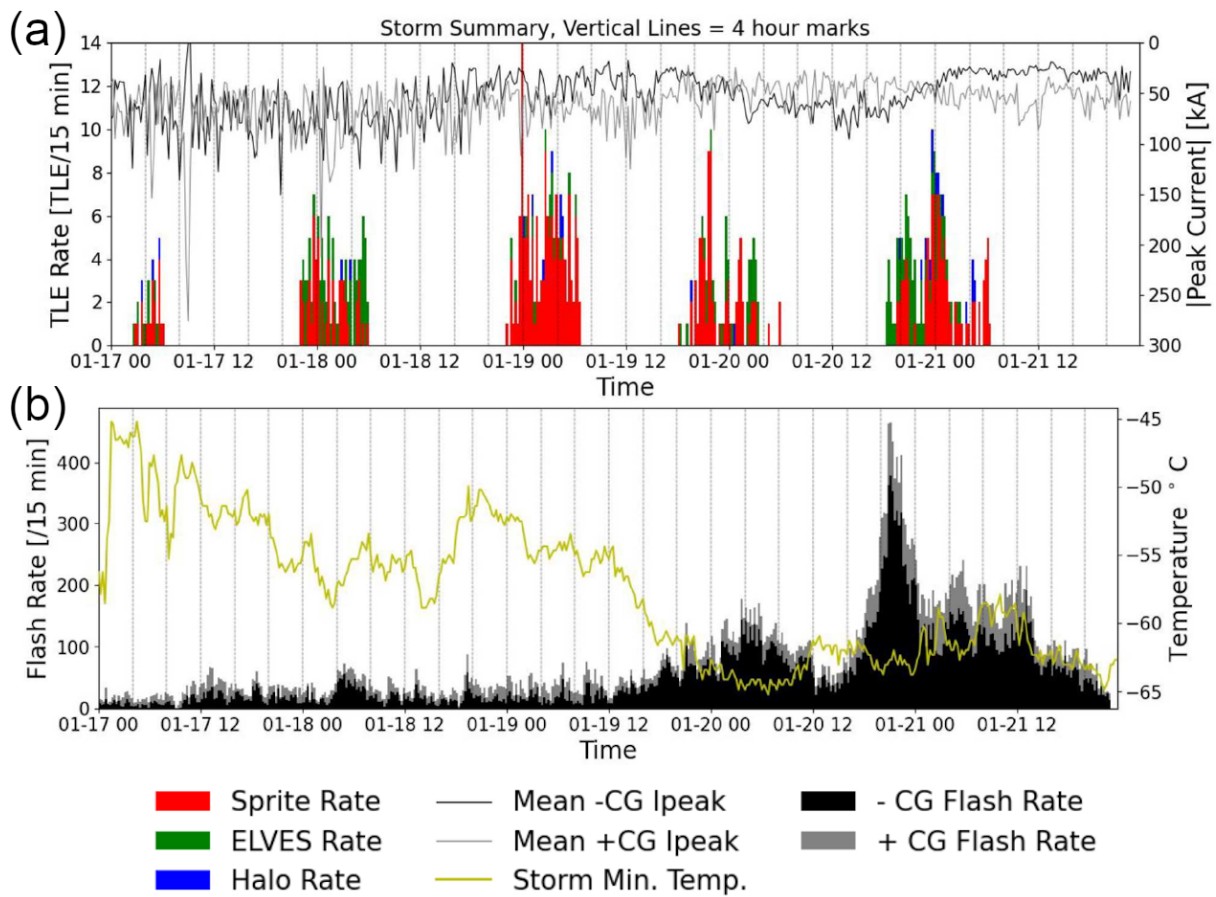

**Figure 11.** TLE recorded at PDM and cloud-to-ground (noted 'CG' here) lightning flash activity between 17 January 0000 UTC and 22 January 0000 UTC (from Gomez Kuri, 2021). (a) TLE rate over 15 minutes intervals (histograms) and mean negative and positive peak current of cloud-to-ground strokes in the same time intervals (black and grey lines, respectively). (b) cloud-to-ground flash rates (negative and positive with black and grey histograms, respectively) and the minimum cloud top temperature over the area of study over 15 minutes (green line). The minimum cloud top temperature is issued from the thermic infrared channel of the Spinning Enhanced Visible and InfraRed Imager radiometer on board the Meteosat Second Generation thanks to the French AERIS/ICARE Data and Services Center.

## 6.2 Instrumental and methodological validation

The rich panel of instruments and infrastructures of both sites makes P2OA a hosting platform suitable for research groups to test new instruments or sensors, which can be compared to reference measurements of the permanent instrumentation. Here we give two examples of such experimentations.

### 6.2.1 Instrumented RPAS and balloons experimentations

With its open spaces (70 ha) mainly composed of prairies, its instruments on atmospheric boundary layer dynamics and its infrastructure (hangar, storage buildings, mechanical and electrical workshops, meeting and working rooms, lodging), CRA is particularly appropriate for balloons and RPAS operations, and airborne instrumentation tests. In total, about 30 test campaigns involving light aircrafts and airborne instruments have been hosted at P2OA since 2015, either with a tethered balloon or with RPAS (free radiosounding balloons are not counted here). In the context of aerial activities and regulation, a Temporary Regulated Area is activated for those operations when needed. The goal of the field campaign is either to test the flying vector, the fly strategy or to test new sensors. Thus, P2OA was the test-site of RPAS and sensors pre-campaigns before international or national field experiments like BACCHUS (Calmer et al., 2019), NEPHELAE (Hattenberger et al., 2022), EUREC4A (Elucidating the role of cloud-circulation coupling in climate, Maury et al., 2023). Strategy of fleet flying (Hattenberger et al., 2022), catapult take off and thread landing have been tested. Sensors for turbulence measurements (Calmer et al., 2018; Alaoui-Sosse et al., 2019, 2022) have been validated based on the P2OA 60-m tower turbulence measurements.

Earlier during BLLAST, several new observational devices or methodologies had been tested or enabled : a technique of frequently-launched radiosoundings with line-cutting system and re-usable probes (Legain et al., 2013), the SUMO RPAS, which is a light tool for frequent profiling of meteorological variables (Reuder et al., 2016) and turbulence (Båserud et al., 2016), a turbulence probe onboard a tethered balloon (Canut et al., 2016), a method for estimating heat fluxes based on frequent profiling of the atmosphere (Båserud et al., 2020).

### 6.2.2 Improvement of airborne gamma-ray technique for radiological surveys and environmental applications

Gamma-ray spectrometry allows the identification and quantification of natural (U- and Th-decay products, $^{40}$K) and artificial (e.g., $^{137}$Cs) radionuclides in the environment. Monitoring their time and space variations offers the possibility to study the environmental factors responsible for these variations, such as soil humidity content and vertical profile (which have an effect on gamma-ray attenuation), surface-atmosphere gas exchange, atmospheric boundary layer dynamics and synoptic transport (which modulate radon flux and atmospheric concentration), dry and wet aerosols' scavenging (which affects the vertical distribution of atmospheric radon decay products) and migration of radionuclides in soils. A suite of instruments dedicated to these studies has been deployed at CRA by CEA/DAM and IRAP since 2018-2019: a 20L NaI(Tl) RSX-5 spectrometer mounted on top of CRA's 60 m tower, atmospheric and soil radon monitors (3 AlphaGuards at 1 m, 30 m and 60 m ; 3 Barasol probes at depths from 15 cm to 90 cm in the ground) and Time Domain Reflectometry soil humidity probes. A specific application consisted in quantifying the influence of environmental parameters (atmospheric radon, soil moisture, cosmic

radiation, atmospheric pressure) on the measurement of natural and artificial radioactivity by airborne gamma-ray spectrometry (Amestoy et al., 2021), in order to improve the survey of sites with nuclear activities or radioactive fallouts, and to characterize the evolution of surface deposits by remote sensing systems such as the Helinuc$^{TM}$ system developed by CEA/DAM (see Fig. 12). A campaign was conducted to monitor the combined evolution of these parameters during fourteen months, combined with several helicopter flights over CRA, and resulting in the correction and validation of the protocol for airborne gamma-ray spectrometry (called PASTHEL) that leads to high measurement precision (Amestoy, 2021). The simulation of a hovering flight by installing the NaI(Tl) spectrometer at 50 m on the 60 m tower made it possible to continuously measure natural radioactivity at a fixed point, focusing on environmental temporal variations. The addition of atmospheric radon detectors made it possible to understand the dynamics of this radioactive gas and its influence on the gamma-ray signal. The installation of Time Domain Reflectometry probes, the study of satellite images and the use of a pluviometer allowed the quantification of the influence of soil moisture and rainfall on the gamma-ray signal. Thus, the effects of atmospheric radon, soil moisture and rainfall could be characterized and taken into account to restore the gamma signal representative of the site's natural radioactivity.

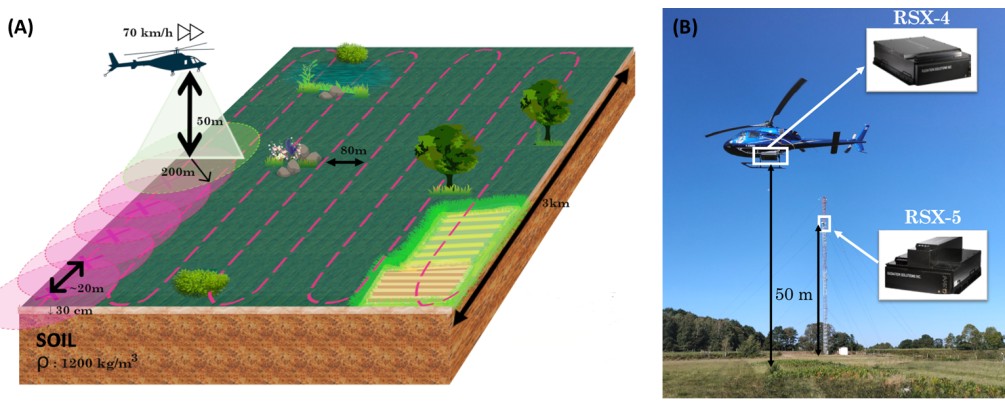

**Figure 12.** (A) Principle of airborne gamma-ray spectrometry. (B) Simultaneous acquisition between the onboard spectrometer (RSX-4) and the one installed on the 60 m tower (RSX-5) at CRA, 8 September 2020.

## 6.3 Evaluation of numerical weather prediction models

### 6.3.1 Atmospheric dynamics variability

As mentioned before, mountain lee waves are typical of the area, generated by southwesterly flow over the Pyrenees. The marked oscillations of the vertical velocity within the whole troposphere that are associated with them, the complex cloud cover system formed, and the impact on temperature, moisture and transport of chemical species, are all difficult to capture in a numerical model, mostly due to the complexity of the terrain itself, and to the challenge of taking the subgrid topography into account.

The measurements of the VHF wind profiler radar at P2OA enable to test the ability of NWP models to simulate the dynamics of the atmosphere in such cases. In particular, the vertical velocity observed by the radar in the mid-troposphere can reach absolute values larger than 2 m s$^{-1}$ during foehn events, while it is usually smaller than 30 cm s$^{-1}$ in other situations (except convective storms). Large vertical velocity variance can thus be observed over a few hours in case of mountain waves in southerly flows, relatively to other typical synoptic situations. The use of a threshold on vertical velocity variance is actually one possible diagnostic of their occurrence (Gueffier et al., 2024). Note that the VHF measurements correspond to rather large scales: a few km horizontally (depending on height, due to beam divergence) and 375 m vertically.

One way to statistically evaluate the NWP models on this aspect is to compare the observed and modeled density energy spectra of the vertical wind component. The two Météo-France NWP models are considered here, from which profiles are continuously extracted at P2OA location, for model evaluations: ARPEGE (Courtier and Geleyn, 1988, 7 km horizontal resolution) and AROME (Seity et al., 2011, 1.3 km horizontal resolution). For AROME, 16 atmospheric columns are averaged around CRA site, and 3 for ARPEGE. Model outputs are extracted at 1 h time resolution. Figure 13 shows the normalized energy density spectra of both the zonal component of the wind (Fig. 13a) and the vertical velocity (Fig. 13b), for a 4-month period of 2018, from January to April. This period was chosen as long enough for this typical comparison example, and without any gap in the observational data. The chosen altitude is close to 3000 m a. s. l., that of the highest peaks at the border. It remains representative of the process, since the whole troposphere, from 1 km or below, up to 8 km or above, coherently oscillates vertically during mountain wave events (Gueffier et al., 2024).

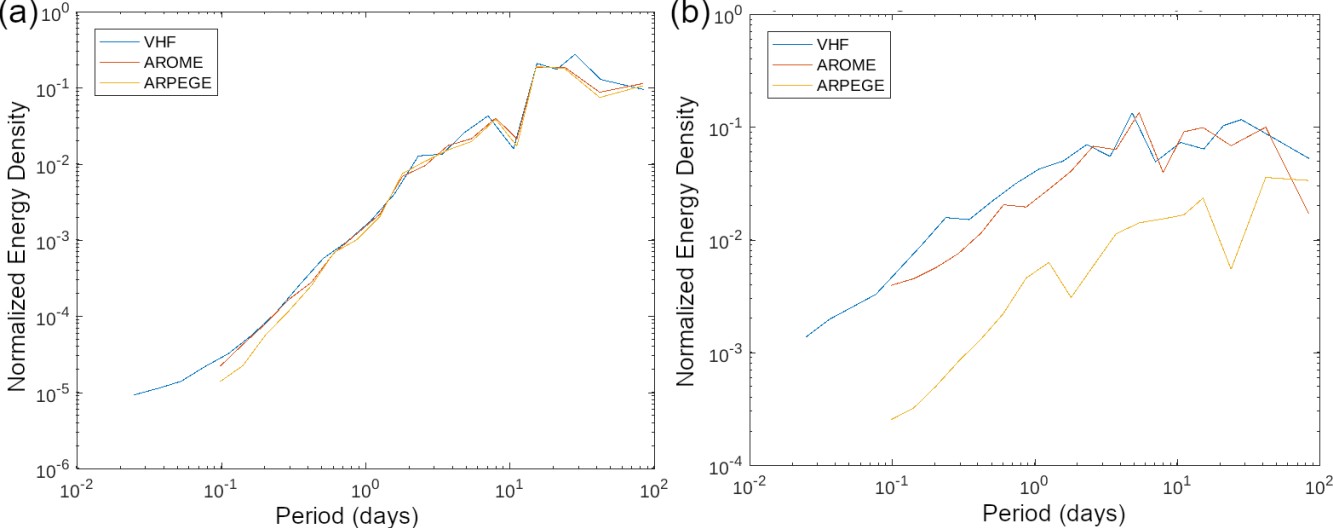

**Figure 13.** Normalized energy density spectra of (a) the zonal component of the wind and (b) the air vertical velocity, at the altitude of 3325 m a. s. l. as observed by P2OA VHF wind profiler radar, modeled by AROME and modeled by ARPEGE, over the 4-month period of January to April 2018.

Figure 13a first shows that the two models correctly represent the variability of the horizontal wind at timescales ranging from 1 h to 1 month. At large synoptic scales, ARPEGE and AROME are very similar, consistently with the forcing of AROME with ARPEGE. They slightly depart from observations around the scale of 30-40 days, which is difficult to interpret. Consistently with their horizontal resolution, ARPEGE slightly departs from the observation at scales smaller than one day, while AROME is remarkably faithful to the observations at scales smaller than one day.

Based on the quality of horizontal wind spectra, one can now consider the vertical velocity variability. Figure 13b reveals how lower is the energy in ARPEGE: the coarser model is not able to represent the variability of the vertical velocity, at any timescale, with energy density about 10 times smaller than observations. Among other processes, it always strongly underestimates the oscillations of the troposphere during foehn or mountain wave events (not shown), which occur around 10 % of the time (Gueffier et al., 2024). The finer model AROME, however, much better represents the vertical velocity variability. It has globally a very satisfying energy density level, even if it slightly underestimates the variance at time scales smaller than 2 days.

This example shows the potential of the 20-year long VHF wind profiler dataset for evaluation of the NWP models and the studies of dynamical processes that are typical of mountain regions.

### 6.3.2 Surface fluxes

As overlined by the MOSAI project (see section 6.1.1), there is a clear need to dig into the representation of surface fluxes in the NWP and climate models. The long-term flux series like those of CRA can now be taken as a reference for their evaluation. Figure 14 shows the difference between the modeled fluxes and the observed fluxes, for both the sensible and the latent heat fluxes. It gives an example only for one year (2021), and with monthly averages, so that inter-annual and intraseasonal variability are worned off here. The overestimation of the sensible and latent heat fluxes by the two models from April to September is consistent with what Couvreux et al. (2016) found during BLLAST experiment in June 2011. One plausible explanation was related to the land-use (with more forest than real land-use in several grid points, especially in ARPEGE) and soil moisture. In specific conditions during BLLAST though, which correspond to a heat wave and very small or negative surface fluxes, the models then underestimated the fluxes. In fall and winter here, we find in average an underestimation of the sensible heat flux by the models, while the latent heat flux remains generally overestimated. The former could be related to the underestimation of the 2 m temperature almost all year long (not shown), with larger negative bias in winter and fall, reaching down to -2.5 °C in average in December 2021 in ARPEGE for example. This reveals the difficulty of such comparisons when the environment (temperature, relative humidity, wind, radiation and clouds, soil conditions...) may itself differ between model and observation, leading to different surface heat fluxes. In this case, it is not possible to conclude on the capability of the models to correctly represent the surface fluxes.

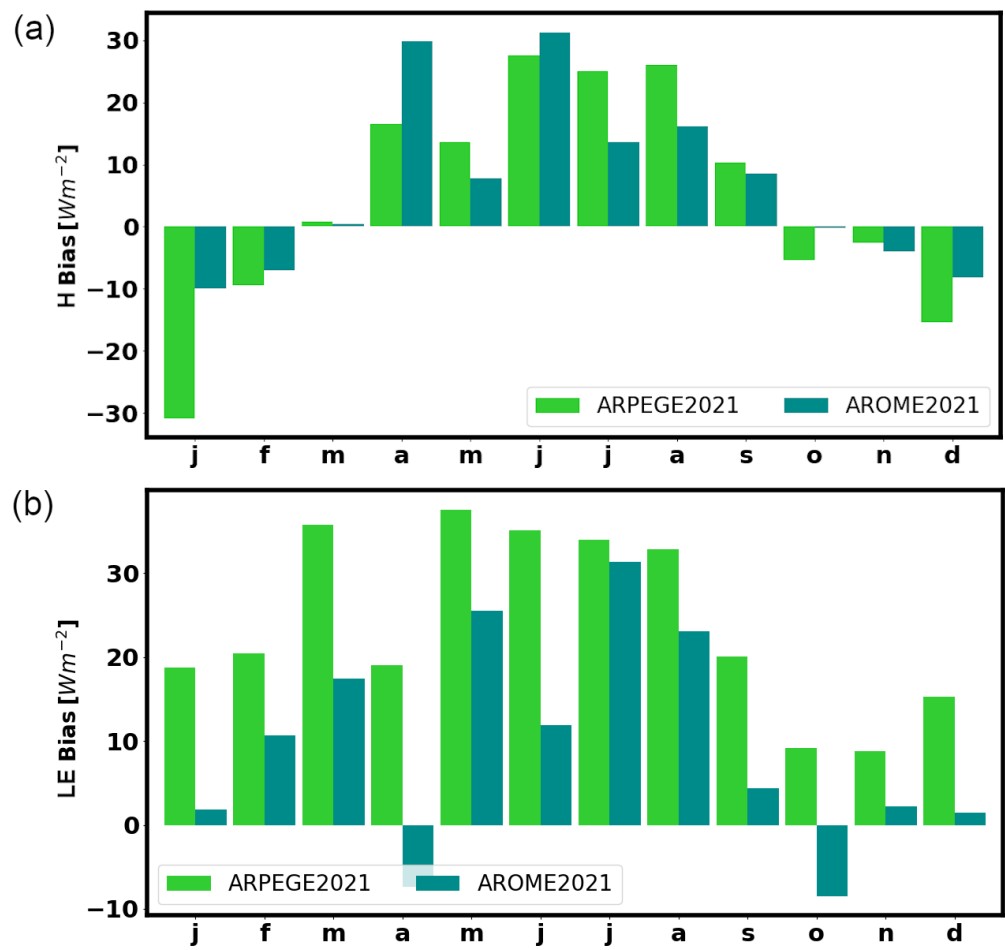

**Figure 14.** Difference between the monthly (a) sensible and (b) latent heat fluxes observed at 30 m at CRA and simulated by ARPEGE and AROME NWP models.

This type of comparison is thus of clear interest as a first step for model evaluation, but is not sufficient to conclude on true
model biases, and their sources. One first need to fully understand the representativity of the measured flux, and the meaning
of the modeled flux on a given grid point. It is, as a second step, important to consider an appropriate way to compare both
together for model evaluation, different such a direct point-to-point comparison. This is the core question of MOSAI project
(Lohou et al., 2023). In its context, Zouzoua et al. (2024) for example propose a new method for model evaluation, based on
supervised neural networks.

### 6.3.3 Validation of a regional climate model on aerosol composition

A one-year sensitivity study was performed at the European scale with the Regional Climate Model (RegCM, Filippo et al.,
2012) over the year 2010. Aerosol radiative forcings and feedbacks are highly dependent of the physical, optical and chemical

properties of aerosols, and of their spatial and temporal distributions. Aerosol sources, transport and sinks performances of RegCM were assessed for the 2010 year over the complex terrain of Pyrenees thanks to PDM aerosol dataset.

Simulated monthly mean aerosol concentrations were evaluated against in situ aerosol concentrations of elemental carbon (EC), organic carbon (OC), aerosol sulfate ($SO_4^{2-}$), calcium ion ($Ca^{2+}$), magnesium ion ($Mg^{2+}$), chlorine ion ($Cl^-$), sodium ion ($Na^+$) from PDM. $Ca^{2+}$ and $Mg^{2+}$ are considered as dust tracers, and $Cl^-$ and $Na^+$ as sea salt (SSLT) tracers. Monthly means of PDM aerosol tracer concentrations were calculated from weekly filter samplings. The weekly sampling integrated volumes of air pumped continuously for 7 days. The chemical analyses performed on these filters were: ion chromatography

analysis for the inorganic fraction of aerosols (World Meteorological Organization Quality Assurance/Science Activity Centre (WMO QA/SAC)[12], for laboratory number 700106); and thermo-optical analysis according to the IMPROVE protocol (Chow et al., 2007) for the organic fraction of aerosols for the year studied. The analytical errors were estimated to 5%.

        In Fig. 15, monthly averages for 2010 are plotted, as well as the monthly averages over the period 2002-2018. Comparing these two trends allows the 2010 variation to be compared to an average variation over a 17-year dataset. The 2010 seasonal

variation lays within one standard deviation around the multi-year average. Only the month of July 2010 shows an average of each chemical compound outside the multi-annual average and associated standard deviation, which is related to the fact that the number of samples contributing to the average is less important and represents specific episodes. The high standard deviations around the multi-annual mean of calcium and magnesium ions (Fig. 15b) shows the occasional occurrence of intense dust episodes over PDM.

The BC and OC seasonal variability is well reproduced by the model as the modeled seasonal variation of BC and OC compounds is comparable to the seasonal variation of measured EC and OC. The model seems to well reproduce the temporal occurrence of regional scale biomass burnings and secondary organic aerosol during the summer period (Fig. 15a). Only the magnitudes of these annual trends are not comparable, the model largely underestimating the concentrations. This is related to the overestimated dry deposition for carbonaceous aerosol parameterized in the model.

Modeled sulfate concentrations show a temporal evolution and an intensity of comparable in order of magnitude with sulfate measurements (Fig. 15c). Figure 15b shows that the model overestimates the dust aerosol concentrations all along the year with a large magnitude. Tsikerdekis et al. (2017) showed through a sensitivity study, that RegCM overestimates dust emission fluxes. The seasonal variation seems to present little consistancy with the measurement during the months of March and April. Concerning sea-salt concentrations (Fig. 15d), the modeled concentrations show an annual variation that does not agree with

the measurement of marine influence tracers at PDM. This reveals again the problem of overestimation of the marine aerosol sources parameterized in the model according to the intensity of the wind fields in cyclonic season over the oceanic area.

# 7   Conclusions

To conclude, we have shown that a broad spectrum of scientific questions and applications can be addressed based on the long-term data set collected at P2OA, due to the rich set of instruments for meteorological dynamics, atmospheric composition

---

[12]http://www.qasac-americas.org/

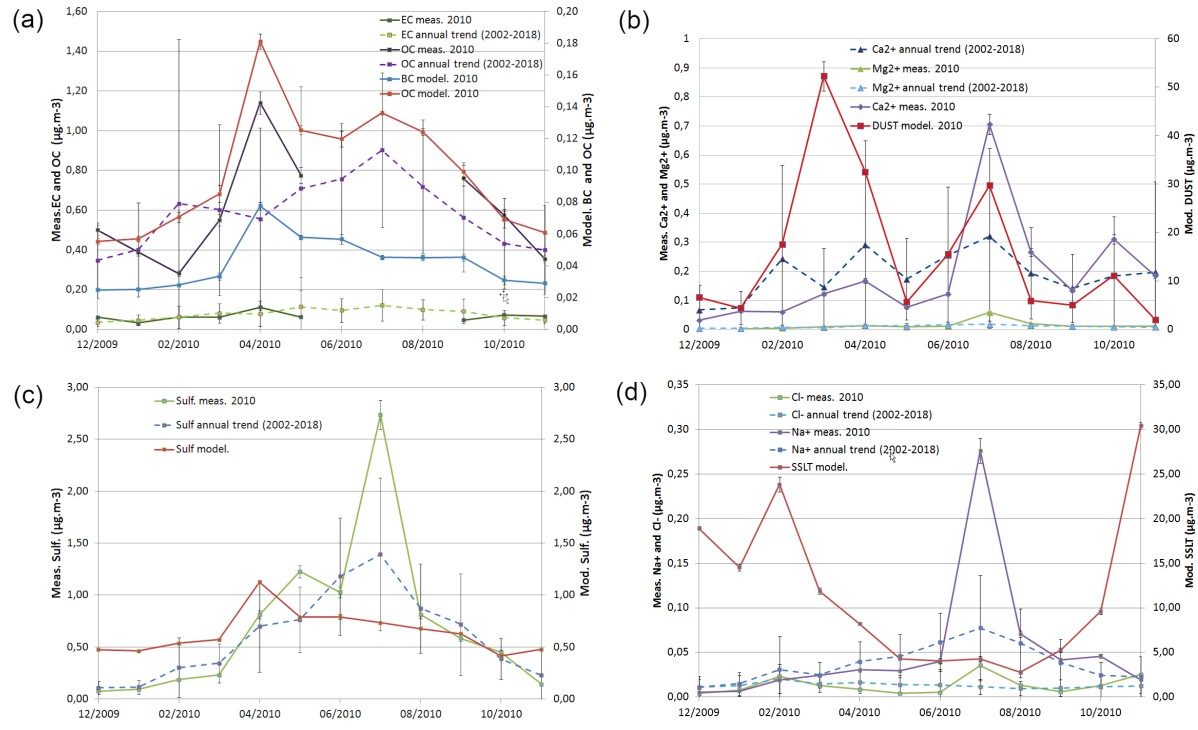

**Figure 15.** Annual trends of monthly mean mass concentrations ($\mu$g m$^{-3}$) of aerosol compounds modelized with RegCM in the Pyrenean region, over the year 2010 with horizontal spatial resolution of 35 km $\times$ 35 km, compared to monthly mean concentrations measured in situ at PDM. (a) Modeled BC and OC mixing ratios with measured EC and OC. (b) Modeled dust mixing ratio with $Ca^{2+}$ and $Mg^{2+}$ measured as main tracers of dust influence. (c) Modeled $SO_4^{2-}$ mixing ratio with measured sulfate concentrations. (d) Modeled SSLT mixing ratio with $Na^+$ and $Cl^-$ measured concentration as main tracers of marine influence.

or electricity. Emphasis is made on specific expertises like atmospheric boundary layer dynamics, trace gas transport, aerosol chemical and physical properties, or transient luminous events. Most continously operated instruments are connected to a French or international network, for weather forecast service, climate and air quality monitoring, or atmospheric process studies.

From the point of view of air composition, the location of P2OA near the Atlantic Ocean makes it weakly influenced by
880 human activity on continental Europe. Further, its implementation near the Pyrenees makes it specific for complex terrain studies. The coupling of the high altitude site at PDM and the plain site at CRA favours the analysis of orographically-forced regimes and their impact on exchange and transport of trace species.

With such a set of instruments and the capability of its infrastructure (lodging, workshops, meeting and conference room, etc.), P2OA is also a well-equipped hosting facility. CRA is especially suited to operate light RPAS and balloons, and PDM
for experiments in high altitude. But beyond these specificities, a broad spectrum of instrumental tests or field campaigns took place at P2OA. P2OA is also a favourable and recognized place for educational training in atmospheric research: 8 to 10

undergraduate or master level trainings are yearly organized, based on micro field campaigns (radiosoundings, surface energy balance station, etc.), and the use of the permanent instruments.

In the context of the main science topics addressed at P2OA and illustrated before, perspectives of new instrumentation is foreseen:

– A new high resolution Doppler lidar should be installed on CRA site by the end of 2024, which will be complementary of all other devices for the description of the atmospheric boundary layer dynamics. Its fine temporal and spatial resolution should give more turbulence statistics in clear air, and open new horizons for the turbulence retrieval observational techniques.

– As coming contribution to the European ACTRIS infrastructure, two monitoring systems of atmospheric short-lived reactive gases will be installed at PDM in 2024: a proton-transfer-reaction – time-of-flight mass spectrometer for measuring a selection of volatile organic compounds, and a $NO/NO_2$ monitoring system. These instruments will complement existing aerosol and gas measurements at PDM, as the targeted species are chemical precursors of ozone and secondary aerosols. They will also allow for a much better characterization of emission sources affecting the sampled air masses:

biomass burning (from agriculture or forest fires), anthropogenic benzene sources, oceanic influence, etc.

*Data availability.* All the dataset is available at https://p2oa.aeris-data.fr/catalogue/. Part of the data is also available from https://www.actris.fr/actris-fr-data-centre/, and from https://www.aeris-data.fr/catalogue/.

## Appendix A: Instrumentation of the 60 m tower

**Table A1.** Instruments installed on the 60 m tower of P2OA-CRA, listed by height, with corresponding measured variables and sampling frequency.

| Height | Sensor | Main variable(s) | Frequency |
|---|---|---|---|
| 2 m | Barometer | Pressure | 0.1 Hz |
| | Rain gauge | Rainfall | 0.1 Hz |
| | HMP45 | Temperature, Humidity | 0.1 Hz |
| | Flux plates (3) | Ground heat flux | 0.1 Hz |
| 15 m | HMP45 | Temperature, Humidity | 0.1 Hz |
| | wind vane | Wind direction | 0.1 Hz |
| | wind anemometer | Wind speed | 0.1 Hz |
| 30 m | HMP45 | Temperature, Humidity | 1 Hz |
| | Sonic anemoter CSAT3 | 3D wind, Virtual temperature | 10 Hz |
| | Licor LI7500 | Water vapour and $CO_2$ concentration | 10 Hz |
| 45 m | HMP45 | Temperature, Humidity | 1 Hz |
| | Sonic anemoter GILL | 3D wind, Virtual temperature | 10 Hz |
| | wind vane | Wind direction | 0.1 Hz |
| | wind anemometer | Wind speed | 0.1 Hz |
| 60 m | Radiometer CNR1 | Up. & Down. SW & LW radiation | 1 Hz |
| | HMP45 | Temperature, Humidity | 1 Hz |
| | Sonic anemoter CSAT3 | 3D wind, Virtual temperature | 10 Hz |

**Appendix B: Glossary of acronyms**

**Table C1.** List of acronyms or names, for research infrastructures, networks, databases or algorithms

| Acronym | Definition | URL |
|---|---|---|
| ACTRIS | Aerosol, Clouds and Trace gases Research InfraStructure | https://www.actris.eu/ |
| AERIS | *Data and Services for the Atmosphere* | https://www.aeris-data.fr/en/welcome-2/ |
| ARM | Atmospheric Radiation Measurements | https://www.arm.gov/ |
| AROME | *Meteo-France small scale numerical prediction model* | https://www.umr-cnrm.fr/spip.php?article120&lang=en |
| ARPEGE | *Meteo-France global numerical weather prediction model* | http://www.umr-cnrm.fr/spip.php?article121&lang=en |
| EC, BC and OC | Elemental Carbon, Black Carbon and Organic Carbon | |
| CALOTRITON | *Algorithm for CBL depth retrieval from radar wind profiler* | |
| CBL | Convective Boundary Layer | |
| CNRS | Centre National de la Recherche Scientifique | https://cnrs.zoom.us/ |
| CO-PDD | Cézeaux-Aulnat-Opme-Puy De Dôme | https://opgc.uca.fr/co-pdd |
| DESMAN | *Algorithm for wind vector retrieval from radar wind prolifer data* | |
| ELIFAN | *Algorithm for cloud fraction estimation from sky imagers* | |
| EUMETNET | European MEterological services NETwork | https://www.eumetnet.eu/ |
| EUSAAR | *Thermo-optical protocol for elemental & organic carbon retrieval* | |
| E-Profile | EUMETNET Profile | https://www.eumetnet.eu/activities/observations-programme/current-activities/e-profile/ |
| GAW | Global Atmospheric Watch | https://gaw/ |
| GMOS | Global Mercury Observation System | https://gmos.aeris-data.fr/ |
| GOS$^4$M | Global Observation System for Mercury | http://www.gos4m.org/ |
| ICOS | Integrated Carbon Observation System | https://www.icos-cp.eu/ |
| iGOS$^4$M | Online database of mercury stable isotopes | http://igos4m.com/ |
| INSU | National Institute of Universe Sciences | https://www.insu.cnrs.fr/en |
| IRSN | Institute for Radioprotection and Nuclear Safety | https://en.irsn.fr/ |
| Linet | Lightning Detection Network | https://www.nowcast.de/en/solutions/linet-systems/ |
| NDACC | Network for the Detection of Atmospheric Composition Change | Lightning Detection Network |
| NWP | Numerical Weather Prediction | |
| OHP | Observatory of Haute Provence | https://ohp-geo.obs-hp.fr/ |
| OPAR | Observatory of Atmospheric Physics at La Réunion | https://lacy.univ-reunion.fr/observations/observatoire-du-maido |
| P2OA | Pyrenean Platform for Observation of the Atmosphere | https://p2oa.aeris-data.fr/ |
| ReNAG | GNSS national network | https://www.osug.fr/missions/observation/terre-solide/renag-gnss-permanent/ |
| ReOBS | *New approach to synthesize long-term multi-variable dataset* | https://reobs.aeris-data.fr/ |
| Ro5 | Ring of Five | https://www.iur-uir.org/en/pro/task-groups/id-22-ring-of-five-task-group |
| RPAS | Remotely Piloted Airplane System | |
| SIRTA | Instrumented Site for Remote Sensing Atmospheric Reserch | https://sirta.ipsl.fr/fr/home-fr-2/ |
| StatIC | Infoclimat STAtions network | https://www.infoclimat.fr/stations/static.php |
| STRAT-Finder | *Algorithm for CBL depth retrieval from lidar and ceilometer* | https://gitlab.in2p3.fr/ipsl/sirta/mld/stratfinder/stratfinder |
| Teleray | *Gamma dose rate alert national network* | https://teleray.irsn.fr/#mappage |
| TLE | Transient Luminous Events | |
| TOPROF | Towards Operational ground based PROFiling | http://www.toprof.imaa.cnr.it/ |
| UFOCapture | Time Shifted Motion Capture Software for High Definition images | https://sonotaco.com/soft/e_index.html |
| UHF | Ultra High Frequency | |
| VHF | Very High Frequency | |

*Author contributions.* Marie Lothon and François Gheusi, as scientific coordinators of P2OA, are the main redactors of this article. P2OA principal investigators are contributing authors, notably for sections dealing on their specific topics: Fabienne Lohou for the meterological and flux ground stations; François Gheusi for atmospheric composition; Véronique Pont for aerosol instrumentation and studies; Marie Lothon, Bernard Campistron, Frédérique Saïd for the wind profiler radars; Marie Lothon for the sky imager and ceilometer; Jeroen Sonke for mercury and microplastics; Serge Soula for atmospheric electricity; Corinne Jambert for measurements of NOx and volatile organic compounds; Michel Ramonet for the Greenhouse Gas monitoring; Pierre Bosser for the GNSS antennas; Pierre-Yves Meslin and Julien Amestoy for gamma spectroscopy; Olivier Masson and Romain Vidal for radionucleide activity. Solène Derrien is responsible for CRA instrumentation and data management. Solène Derrien, Emmanuel Leclerc, Antoine Vial, Yannick Bezombes, Eric Gardrat, Gilles Athier operate and maintain P2OA instrumentation and ensure data quality control. Damien Boulanger, Nicolas Pascal, Renaud Bodichon, Yves Meyerfeld contribute or have contributed to the data production and dissemination, in the context of AERIS data service. Bernard Campistron is the author of DESMAN algorithm, Alban Philibert is the author of the boundary layer height retrieval CALOTRITON algorithm (Philibert et al., 2024) and other data process algorithms, he proveided the convective boundary layer height composite; Gilles Athier is the author of the in-cloud index algorithm. Eric Pique, Jean-Bernard Estrampes, Fabienne Guesdon, Felix Starck, Guillaume Bret, Laurent Cabanas and Erwan Bargain contribute or have contributed to the technical maintenance of P2OA site and instrumentation. Jérémy Gueffier provided the monthly composite diurnal cycles of atmospheric composition at PDM. Guylaine Canut has made the comparison between the AROME and ARPEGE operational numerical weather predicting models with P2OA flux measurements. Zaida Gomez Kuri has made the analysis of the Transient Luminescent Events shown in this article. All coauthors provided internal reviews.

*Competing interests.* The authors declare no competing interests.

*Acknowledgements.* P2OA facilities and staff are funded and supported by the University Paul Sabatier Toulouse 3, France, and CNRS (Centre National de la Recherche Scientifique). P2OA is part of the national research infrastructure ACTRIS-France.

Several P2OA dataset are related to ACTRIS-Fr research infrastructure (https://www.actris.fr/). Associated products and services are maintained by the French national center for Atmospheric data and services AERIS. ACTRIS data policy can be found here: https://www.actris.eu/sites/default/files/Documents/ACTRIS%20PPP/Deliverables/ACTRIS%20PPP_WP2_D2.3_ACTRIS%20Data%20policy.pdf.

We thank OMP/SEDOO, IPSL/ESPRI, and ICARE data centers for their contribution to AERIS and P2OA data services. We also specially thank Andry Andriatiana for his involvment in P2OA data base and web site.

Rénag is member of the Research Infrastructure (RI) Résif-Epos, managed by CNRS-Insu. Résif-Epos is inscribed on the roadmap of the Ministry of Higher Education, Research and Innovation, the Résif-Epos IR is a consortium of eighteen French research organizations and institutions. Résif-Epos benefits from the support of the Ministry of Ecological Transition.

We thank Eric Bazile and Yann Seity from Météo-France/CNRM for providing the AROME and ARPEGE operational model outputs, and for their expertise, and Météo-France national service for PDM and CRA synoptic station data.

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
