# Peer review of "The Pyrenean Platform for Observation of the Atmosphere: Site, long-term dataset and science"

_Atmospheric Measurement Techniques, 2024_

## Referee Comment (RC2)

[referee-annotated manuscript omitted]

---

## Author Response (AR1)

**Reply to Reviewer 2**

*We thank the reviewer for her/his careful reading of the manusript, and for all the useful comments which will help improving the manuscript.*

**Major remarks**

1)   Line 10 : It seems to me that the research infrastructures and the main networks in which P2OA is involved should be mentioned in the abstract.

*It is right that this point should be mentioned in the abstract. This has been done in the revised version (see lines 12-14 of the revised manuscript).*

2)   Line 18 : « it is one of the five national multi-instrumented sites" : Most of them are mentioned later in the text, but I think that they all could be mentioned here.

*We have indeed listed the 4 other French INSU instrumented site there in the revised version (see lines 24-28 of the revised manuscript, or lines 25-29 of the visible-corrections-manuscript) .*

3)   Lines 25-30 : It seems not necessary to me to mention here the polar or marine stations whose scientific themes are far from P2OA.

*Yes, P2OA belongs to the most common category of continental non-polar sites. But we wished to remind the large diversity of ground-based sites in the world, including polar sites. We prefered to keep this example.*

Scientific topics are not so far, since atmospheric science is done as well at those sites, like pollution survey and boundary layer dynamics.

4)   Table 1  : The table indicates that greenhouse gas measurements are in the framework of ICOS-FR, but not ICOS-EU. Is this an error or is the data really not transmitted to the European level of ICOS? For what reason ?

*There is no error. Only few selected stations of the French national ICOS network have the European ICOS label (and access to the European central support facilities), but neither PDM nor CRA belong yet to those. It was initially planned to integrate these stations in ICOS as associated stations, a status recognized by the ICOS-ecosystem network but not yet by the Atmosphere component of ICOS. Discussions are continuing with the French Ministry of Research to extend the*

*number of national class 2 labelled stations. That said, it is important to note that both the CRA and PDM stations, supported by national funding, contribute nevertheless to the European datasets compiled by ICOS-ERIC and distributed by the Carbon Portal:*
*CO2 : https://doi.org/10.18160/FSS8-53NX*
*CH4 : https://doi.org/10.18160/9B66-SQM1*
*Those datasets are used in several studies by Saunois et al. 2020, Thompson et al. 2020, Ramonet et al. 2020, Thompson et al. 2021, He et al. 2023, Cristofanelli et al., 2023. For this reason, ICOS has been added as international network including P2OA data in the last column of Table 1 for the greenhouse gases measurements at PDM and CRA.*

*The data dissemination is also ensured by the French national infrastructure AERIS handling atmospheric research data (open to any user worldwide).*

5)  Table 1 : The Table and the article in general contain a lot of acronyms. Some of them are not necessarily well known (TLE, GNSS, VLF, LPR GEM, GOM, etc.). It would be useful for the reader that all the acronyms be defined (potentially in an appendix section).
*We have turned into full text  acronyms that were used only once or twice in the revised manuscript (GEM, GOM, PBM, SEB, GET, GHG, CRDS, ANSTO, SMPS, FOV, ECAC, CiGas, EUCOS, GFS, WDCGG, WOUDC, WDCRG, WDCA, NILU, PVC, PET, MCS, CG , CTT, TDR, PRT-TOF-MS, VLF, LPR), and listed all other acronyms in a glossary appendix, with names and url of web site (Table C1 of appendix B).*

*Note that we did not find « VLF » or « LPR » in the manuscript.*

6)  Table 1 : Add references (scientific articles or web links to algorithms if they exist (5th column) and to networks and research infrastructure (6th and 7th columns).

*We have completed the references or associated url for algorithms and networks, through the text and Appendix B.*

7)  Table 1 : Some variables are within the framework of several networks or Research Infrastructures (example ACTRIS/NDACC, ACTRIS/GAW, ACTRIS/GMOS). Would it be possible to explain the articulation or link between these networks or RI?

*The description of the international network ecosystem is not directly in the scope of this manuscript, which is more specifically dedicated to the P2OA. However, we have added a few lines about the articulation between ACTRIS-Fr and the networks that are connected to it (see lines 319-322 of the revised manuscript, or 322-325 of the visible-corrections-manuscript) :*

*« By construction, ACTRIS-Fr is thus a convergence point of many networks, which is beneficial in both ways: the infrastucture helps the involved sites in maintaining there instrumentation and monitoring, in the data dissemination, brings national scientific research*

*dynamics, etc...*
*and the network brings specific scientific questions, dynamics and tools of a european or international community, etc... »*

8)    Section 1 : The introduction is almost exclusively oriented towards the variables and themes of ACTRIS. Indeed, Table 1 shows that the majority of variables on the two sites are ACTRIS variables, but ICOS and IRSN are also part of the instrumental setup, without being mentioned at all in the introduction. I think it would be necessary to broaden the introduction scope to all the networks and IRs whose data is produced at P2OA.

*It is right that we stressed a lot on ACTRIS and ACTRIS-Fr (since those infrastructures play a key role for P2OA, and P2OA is an important part of ACTRIS-Fr), while P2OA topics span a much broader spectrum than ACTRIS. We insisted more on this broad spectrum in the revised version, removing the ACTRIS angle of view that the introduction initially seemed to start with, insisting more on this large spectrum, and ending with the involvment of P2OA in ACTRIS, ICOS and several atmospheric survey networks.*

*See lines 41-60 of the revised manuscript, or 42-62 of the visible-corrections-manuscript)   revised manuscript.*

9)    Section 5.2 : This section discusses the seasonal variability. Figure of temperature, vapor and shortwave radiation at CRA. It seems that figure 3 to 6 are only made with CRA data, but meteorological measurements are also performed at PDM. Their use could be useful to characterize the meteorological variability of the entire instrumented site and not just CRA? If the section is limited to CRA this should appear in the section title (not only seasonal variability, but Seasonal variability + of what + where)

*This is true. We first thought that for the caracterisation of meteorological variables, the data from CRA were sufficient, and representative of the regional scale. Indeed, as mountain-top observatory, the PDM is a very specific spot with  buildings surrounding the meteorological station, and measurements cannot be operated with best standards. Meteo-France has classified its PDM station in class #5 for temperature and humidity, which is the poorest level, due to the environment of the station (not flat, influence of close buildings).*

*The wind measured at PDM should not be used without  care, because it is measured in the lee of a large and 150 m antenna in case of northwesterly flow  (see Hulin et al. 2019). For this reason, using the wind measured by the VHF at the altitude of the PDM is more representative of  wind at this altitude, and of the synoptic flow in the lower free-troposphere.*

*Temperature and moidture may also be affected by this lack of representativity, but less affected than wind.*

*Of course, measurements made at this altitude for decades remain extremely useful and interesting.*

*With all this in mind, and encouraged by the relevant suggestion of Rev#2, we eventually decided to add temperature and water vapour mixing ratio measured at PDM in Figure 3. We discuss them in Fig. 3 and lines 436-451 of the revised manuscript ( or 442-457 of the visible-corrections-manuscript). Presenting those variables enables us to show the amplitudes of the temperature seasonal and diurnal cycles at PDM relatively to CRA. Temperature has a larger seasonal amplitude at PDM, and water vapour mixing ratio has it smaller. From April to October, water vapour mixing ratio (temperature) shows a larger (smaller) diurnal amplitude at PDM. This is likely due to the occurrence of deep CBL at midday during this period, including the PDM top, but leaving the PDM top in the free troposphere during the night. To the contrary, in winter months, PDM summit is always in the free troposphere, with no or weak diurnal cycle (and anyway less marked than those still observed at CRA).*

10) Section 5.3 : This section discusses the atmospheric composition seems only limited to PDM (contrary to section 5.2 limited to CRA). Table 1 indicates that there are also atmospheric composition measurements at CRA. Is it possible to broaden the context of atmospheric composition to the entire instrumented site, including CRA?

*Although the covered measurement period is not exactly the same for all variables, we have extended this analysis to the CRA site (Figure 8 of new manuscript) and discussed the new figures, which represent a rural plain site (see section 5.4).*

11) Figure 7 : With the period covered is limited to 2015-2019? Is it the longer period during which variables are all available ?

*This period correspond to that studied in the PhD work by J. Gueffier (2023, https://theses.fr/2023TOU30385) and in Gueffier et al. 2024. We used here the same data set, with the benefit of a good data coverage for all variables, synchronization at a common hourly time base, and an attentive data quality control. A mention to this has been inserted in the text of Section 5.4 (see lines 523-526 of revised manuscript lines 530-533 of the visible-corrections-manuscript).*

12) Section 5 : A subsection "long term trends" should be welcome to complete the seasonal variability, atmospheric composition …etc. Table 1 indicates that the P, T, Humidity and wind variables have an indicated start time period of 1882. Is historical dataset from the 19th century available anywhere? Do they show a very long term trend?

*We first abandoned the idea to show such a trend analysis, for different reasons: difficulty of homogeneity in the long historical series (and despite of the big efforts of Dessens 1995), perspective of a future work and publication on those long term data, after an important work of numerisation of the historical data, wish to cross greenhouse gas measurements and cloud observation with the meteorological data.*

*However, it is true that it would be a pity not to present this long historical series of observation of*

*temperature at Pic du Midi in this presentation paper.*

*So we have presented the temperature evolution at both sites since the start of the temperature measurements in the revised version. This was done in a new section (5.3 in the revised version) and new figure (Fig. 7).*

13) Section 6 : The subsections present scientific results obtained with P2OA data, but some were also performed within the framework of research infrastructures (ACTRIS, ICOS, ...) or networks (GAW, IRSN, ...). These networks should be mentioned in the text. For example, are the results presented in Section 6.2.2 within the framework of the IRSN or not? If not could one IRSN scientific result bee highlighted ?

*We have verified this point, and added mention of IRSN or AERIS when appropriate.*

*See lines 676 & 723 of therevised manuscript (or 684 & 733 of the visible-corrections-manuscript).*

14) Section 6 : There does not seem to be any highlighted scientific result based on ICOS measurements in section 6? why ?

*The P2OA sites are part of the national ICOS-France network, but not yet labelled as European ICOS-ERIC stations. They both contribute to the European datasets compiled annually by ICOS for $CO_2$ (https://doi.org/10.18160/FSS8-53NX) and $CH_4$ (https://doi.org/10.18160/9B66-SQM1). In particular, the PDM station was at the heart of the evaluation of the method for spike detection algorithm which is now applied to all the stations in the ICOS network (El Yazidi et al., 2018; Tenkanen et al., 2021 ; Cristofanelli et al., 2023). P2OA data have also been used to study the impact of European droughts on the $CO_2$ concentrations and fluxes (Ramonet et al., 2020; Thompson et al., 2020; Jiang et al., 2023), as well as to study global and European methane balances (Saunois et al., 2019; Szenasi et al., 2022; Thompson et al., 2021).*
*We have added those studies in a new paragraph « Greenhouse gases » in the revised version of section 6.1.2 (see lines 682-689 of the revised manuscript, or 692-699 of the visible-corrections-manuscript).*

*$CO_2$ and $CH_4$ ICOS data were also used in several studies cited in the article (Hulin et al, 2019 ; Roman-Cascon et al., 2019 ; Gueffier, 2023 ; Gueffier et al, 2024), as well as in the present article itself (Section 5.3 and 6.1.2). This is now more precisely mentioned in the text (see lines 522 of the revised manuscript, of 529 of the visible-corrections-manuscript).*

15) Figure 8: the horizontal axis may seem mysterious to a non-specialist reader. Is it possible to explain a little bit its meaning and the unity used?

*In this figure, time is represented with normalized dimensionless time, which is common in turbulence decay studies. It was already explained in the caption, but this was probably not clear*

*enough.*

*This dimensionless time is based on the period starting at the time of maximum surface buoyancy flux during midday, and ending at the time when surface buoyancy flux gets to zero, later in the afternoon.*

*We now have explained it more precisely in the text, and added an equation for this.*
*We also added milestones on the figure itself, indicated normalized time when buoyancy flux is maximum, and when it is zero.*

*See Figure 9 and lines 600-605 of revised manuscript (or lines 609-613 of the visible-corrections-manuscript).*

**Minor points**

1)   Figure 1 : Top part: add a color bar which indicates the altitude

*There is no available legend on the IGN Geoportail from which we have obtained this relief map ; but we have added indication of altitudes in the caption of Fig. 1.*

2)   Figure 2 : the radial scale is not the same (max 12% on a, 14% on b and c). It would be better to put the same scale on the 3 plots (all 14%)

*The wind distribution is slightly less visible when the percentage axis in panel (a) is homogenized with other panels. However we followed Rev#2 suggestion, for homogeneity of all panels (See Fig 2).*

3)   Figure 6: the legend indicates that the variable on the figure is Zi and the text line 449 indicates the figure shows the convective boundary layer (CBL). Is it the same variable ? Need to homogenize between text and figure caption.

*Yes, this was not clear in the manuscript. We have precised this in the text (lines 475 of the revised version, or line 481 of the visible-corrections-manuscript) and in the caption of Figure 6.*

4)   Figures 3, 4, 5, 6, 10 : The site where the data have been obtained (CRA or PDM) has to be mentioned in the caption.

*We have mentioned the site each time in the legends of those figures.*

5)   Line 510 : the web link is in the text, while the web links in section 4.1.5 are in footnotes. It seems preferable to standardize by putting all the links to the websites in footnotes as example (or in appendix).

*We have put this url as a footnote, homogenized all the references to web sites, and moved all the URL references associated to acronyms of network, data base/center or project, to Table C1 in the appendix, which draws the list of acronyms.*

**Reply to Reviewer 3**

*We thank the reviewer for his/her formal feedbacks.*

*We have taken account of them all in the revised manuscript- lines 35, 49, 335 and throughout the text for the mention of the atmospheric boundary layer.*